# Detecting turbulent structures on single Doppler lidar large datasets: an automated classification method for horizontal scans

Ioannis Cheliotis[1], Elsa Dieudonné[1], Hervé Delbarre[1], Anton Sokolov[1], Egor Dmitriev[2], Patrick Augustin[1], Marc Fourmentin[1]

[1] Laboratoire de Physico-Chimie de l'Atmosphère (LPCA), UR 4493, Université du Littoral Côte d'Opale (ULCO), Dunkirk, France

[2] Marchuk Institute of Numerical Mathematics Russian Academy of Sciences, Moscow, Russia

*Correspondence to*: Ioannis Cheliotis (ioannis.cheliotis@univ-littoral.fr)

**Abstract.** Medium-to-large fluctuations and coherent structures (mlf-cs) can be observed using horizontal scans from single Doppler lidar or radar systems. Despite the ability to detect the structures visually on the images, this method would be time-consuming on large datasets, thus limiting the possibilities to perform studies of the structures properties over more than a few days. In order to overcome this problem, an automated classification method was developed, based on the observations recorded by a scanning Doppler lidar (LEOSPHERE WLS100) installed atop a 75-m tower in Paris city centre (France) during a 2-months campaign (September-October 2014). The mlf-cs of the radial wind speed are estimated using the velocity azimuth display method over 4577 quasi-horizontal scans. Three structures types were identified by visual examination of the wind fields: unaligned thermals, rolls and streaks. A learning ensemble of 150 mlf-cs patterns was classified manually relying on *in-situ* and satellite data. The differences between the three types of structures were highlighted by enhancing the contrast of the images and computing four texture parameters (correlation, contrast, homogeneity and energy) that were provided to the supervised machine learning algorithm, namely the quadratic discriminate analysis. The algorithm was able to classify successfully about 91 % of the cases based solely on the texture analysis parameters. The algorithm performed best for the streaks structures with a classification error equivalent to 3.3 %. The trained algorithm applied to the whole scan ensemble detected structures on 54 % of the scans, among which 34 % were coherent structures (rolls, streaks).

## 1. Introduction

Turbulent flows are motions characterized by high unpredictability. Nevertheless, coherent structures are developed in these flows (Tur and Levich, 1992). The principal aspect that determines a coherent structure is the maintenance of the phase-averaged vorticity of the turbulent fluid mass over the spatial extend of the flow structure (Hussain, 1983). The most typical types of coherent structures are presented in the review of Young et al (2002), who classified structures into three characteristic types: turbulent streaks, convective rolls and gravity waves. Several studies have been carried out to examine the effect of the coherent turbulent structures in the dispersion of pollutants by utilizing boundary layer simulations. The results of these studies indicate that the coherent structures can play a significant role in the pollutants' concentrations (Aouizerats et al., 2011; Soldati, 2005). Furthermore, Sandeepan et al., (2013) have demonstrated via simulations that the pollutants' concentrations can alternate from low to high during coherent structures events. It is therefore important to be able to identify structures in the atmosphere and observe them in an efficient and consistent way. The term coherent structures in the aforementioned studies refers exclusively in the atmospheric flow and it is the main focus in this study. This term is also encountered in studies at laboratory scale described as hairpins or packets (Adrian, 2007; Hutchins and Marusic, 2007), but these are out of the scope of this study.

Turbulent streaks are structures aligned with the horizontal wind with alternating stripes of stronger horizontal wind associated with a subsidence and stripes of weaker horizontal wind associated with an ascendance (Khanna and Brasseur, 1998). The high wind shear between the surface layer and the lower planetary boundary layer (PBL) can lead to the formation of the turbulent streaks in the surface layer that may extend to the mixed layer. Neutral or near-neutral stratification favours the formation of streaks, though they may also form during stable and unstable conditions (Khanna and Brasseur, 1998). The physics behind their formation differs as the contribution of buoyancy varies in relation to the atmospheric conditions (Moeng and Sullivan, 1994). Formation, evolution and decay of streaks are rather short, equivalent to several tens of minutes, before they regenerate. The average streak spacing is usually hundreds of meters (Drobinski and Foster, 2003). In the mixed layer, horizontal roll vortices, also known as convective rolls, develop roughly aligned with the mean wind (LeMone, 1972). Favourable conditions for the development and maintenance of convective rolls are the spatial variations of surface-layer heat flux, the low-level wind shear and the relatively homogeneous surface characteristics (Weckwerth and Parsons, 2006). As the rolls rotate in the vertical plane, they generate ascending and descending motions. These motions under convective conditions can form clouds in rows separated by clear sky areas known as cloud streets which is a characteristic visual feature used to identify rolls (Lohou et al., 1998). The rolls usually extend from the surface to the capping inversion with a large variety of horizontal sizes from few kilometers to few tens of kilometers. They are characterized by long lifespan of hours or even days as opposed to the short lifespan of the streaks (Drobinski and Foster, 2003). Young et al (2002) distinguish rolls in narrow mixed-layer rolls, where the ascending air masses are one thermal wide (Weckwerth et al., 1999) and wide mixed-layer rolls, where multiple thermals are grouped within each ascending area (Brümmer, 1999). As Young et al (2002) stated, both types of rolls can be distinguished visually, with the narrow rolls having the form of a "string of pearls" whereas the wide rolls look like a "band of froth".

Remote sensors are exceptionally useful for the identification of coherent structures. Their ability to scan large areas in a short period is advantageous compared to in situ measurements (Kunkel et al., 1980). Lhermitte (1962), Browning & Wexler (1968) were the first to implement the velocity azimuth display (VAD) technique, also known as plan position indicator (PPI) method, using Doppler radars. The PPI technique provides conical scans or even horizontal surface scans with the appropriate combination of elevation and azimuth angles. Kropfli & Kohn in 1978 were able to study horizontal roll structures by using a dual-Doppler radar in order to observe the wind field in the three dimensions. Several studies followed for different type of radars with more efficient configurations (Kelly, 1982; Lohou et al., 1998; Reinking et al., 1981). Weckwerth et al. (1999) were able to study the evolution of horizontal convective rolls by combining Doppler radar observations with meteorological measurements, radiosondes, flight measurements and satellite images. In recent years, various studies have been carried out

by using lidars only. It has been well established that the PPI method can also be applied to Doppler lidars (Cariou et al., 2007; Vasiljević et al., 2016) with the possibility to compute the mean wind profile by using a modified version of the VAD method as it has been demonstrated in the studies of Banta et al., (2002) and Chai et al., (2004). Depending on the selected scanning method of the Doppler lidar, it is possible to observe coherent structures in the atmospheric surface layer (Drobinski et al., 2004) as well as in the mixed-layer (Drobinski et al., 1998). Newsom et al. (2008) and Iwai et al. (2008) introduced the dual-Doppler lidar method and revealed its benefits in the observation of coherent structures. This method was further improved by Träumner et al. (2015) using an optimized dual-Doppler technique. They were able to identify different type of structures including elongated areas resembling turbulent streaks. They combined quantitative characteristics of the coherence such as the integral scales and the anisotropy coefficients, obtained by a two-dimensional autocorrelation algorithm, with the visual observation of the scans. However, the subjective classification by observing the images is a time-consuming approach and non-systematic. Furthermore, the use of two Doppler lidars is limited to the institutes that can afford such a high cost and collaborations on short-term campaigns. A much less expensive approach, and suitable for long periods, is to detect the passage of the structures on sonic anemometer time series. For instance, Barthlott et al. (2007), analysed 10 months of data from a meteorological tower located in the surface layer 20 km south of Paris, France and they observed coherent structures for 36 % of the cases. However, their study is limited to point measurements instead of a larger wind field that it is possible to observe via a lidar.

This study aims to identify the medium-to-large fluctuations and coherent structures (mlf-cs) on single Doppler lidar horizontal scans and develop an automatic classification process based on the combination of texture analysis and a supervised machine learning technique, namely the Quadratic Discriminate Analysis (QDA), in order to handle large datasets. Texture analysis is an effective way to evaluate the distribution of the values within an image (Castellano et al., 2004). It is widely used in various scientific fields in order to classify images, covering meteorology (Alparone et al., 1990), medical studies (Holli et al., 2010) and forestry (Kayitakire et al., 2006). There is a lack of long-term studies of structures based on lidar observations and the aforementioned automatic classification process can stimulate the interest in this research field. More particularly, it could facilitate the statistical analysis of the physical parameters of the structures, e. g. the structure size as a function of the planetary boundary layer (PBL) height. Furthermore, it will enable us to study the transitions between structures and how these are associated to the atmospheric conditions. Finally, the impact of the structures on pollutants' concentrations could be examined for long-term studies under stable and unstable conditions. The classification method relies on the observations of radial wind speed recorded using a scanning Doppler lidar settled atop a 75 m-high tower in the centre of Paris, during a two-month period in late summer/early fall. Section 2 presents the experimental set up of the study. The methodology for the identification and classification of the mlf-cs is demonstrated in Section 3. Subsequently, the results of the classification for the training ensemble as well as for the whole dataset are displayed in Section 4. Finally, the key points of the paper are summarized in Section 5.

## 2.    Experimental set up

A two-month measurement campaign (04/09-06/11/2014) was carried out in order to study the exchange processes of ozone and aerosols in the area in the framework of the VEGILOT [VEGétation et ILOT de chaleur urbain (vegetation & urban heat island)] project in the urban area of Paris (Klein et al., 2019). The Leosphere WLS100 Doppler lidar (www.leosphere.com) with a minimum range of observations at 100 m (Figure 1a) was installed atop a 75 m building in the Jussieu Campus, located in the centre of Paris city (Figure 1b) and was used for wind measurements. Table 1 shows the significant lidar properties during the VEGILOT campaign.

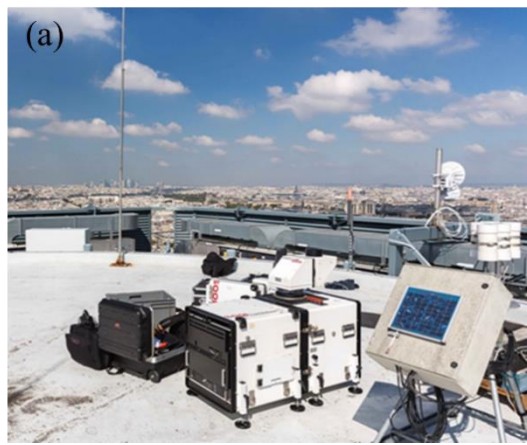 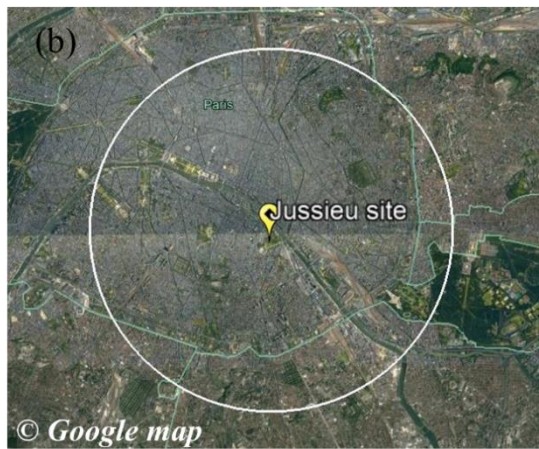

Figure 1: (a) The Doppler lidar installed on the tower roof during the VEGILOT campaign and (b) the measurement site in Paris with the circle of 10 km diameter demonstrating the maximum range of the PPI surface scan (Google earth satellite image).

The Doppler shift frequency between the emitted laser beam and the light backscattered by the aerosols is measured by heterodyne detection associated with Fast Fourier Transform as explained analytically by Cariou et al., (2007). A wind lidar is measuring the radial wind speed i.e. the wind projection along the light beam (counted positive when going away from the lidar). Table 2 showcases the implemented scanning methods during the VEGILOT campaign. For the classification of the mlf-cs, we focused in the current study on the almost horizontal PPI scans (1° elevation angle). During those scans, the lidar emitted beams in azimuth angles from 0° to 360° with a 2° resolution. This scenario was repeated every 18 minutes hence providing 4577 PPI scans during the whole campaign. The duration of each scan was 3 minutes which is sufficiently fast for the observation of coherent structures with a lifespan of several minutes. The maximum range of the scans reached 5 km (see white circle of Figure 1b) with a spatial resolution of 50 m. It is noteworthy that the scanning area covers almost exclusively the urban area of Paris. A city famous for regulating the height of the buildings to not exceed 50 m in its' centre (Saint-Pierre et al., 2010). The ground altitude enclosed by the scanning area mostly ranges between 30 to 60 m with the exception of some hills near the boundaries of the scanning range as it can be seen in Figure 2. It is fundamental for this study to assume that the wind field within the scanning area is homogeneous (see Section 3.1). Due to the 1° elevation, the beam was risen by about 87 m between the central point and the point at the 5 km. It was also important for this study to retrieve observations regarding the vertical wind shear. For this purpose, the Doppler beam swinging (DBS) scanning method was implemented. This method was consisted of four line of sight beams at azimuth angles of 0°, 90°, 180° and 270° with an elevation angle of 75° and it was applied twice. The duration of the four beams emission was approximately 15 seconds.

Table 1: Properties of the lidar used for the observation of mlf-cs

| Doppler lidar (Leosphere WLS100) | |
|---|---|
| Altitude of lidar: | 75 m a.g.l. |
| Minimum range: | 100 m |
| Radial wind speed range: | -30 to 30 m·s⁻¹ |
| Laser wavelength: | 1.543 μm |
| Radial wind accuracy: | ± 0.1 m·s⁻¹ |
| Accumulation time: | 1 sec·beam⁻¹ |

Table 2: Scanning methods selected during VEGILOT

| | Scanning area | Purpose | Elevation & azimuth angle | Scan duration |
|---|---|---|---|---|
| **Plan Position Indicator (PPI)** | Almost horizontal scans near surface | **Identification of structures** | Elevation 1°, azimuth 0 to 360° with 2° resol. | 3 min |
| **Doppler Beam Swinging (DBS)** | Combination of LOS | Identification of low level jet cases | Elevation 75°, azimuth 0°, 90°, 180° & 270° | 2 x 15 sec |

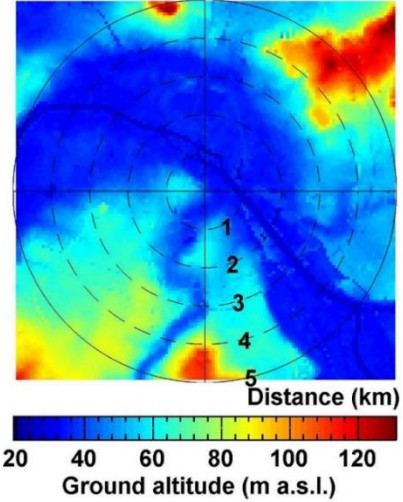

Figure 2: Ground altitude map above sea level for the scanning area in Paris.

## 3.    Preparation of the dataset for the classification

### 3.1    Turbulent radial wind fields

Assuming a homogeneous wind field for horizontal PPI scans, the radial wind measurements $u_r$ taken for the different beams at a given distance from the lidar should follow a cosine function of the azimuth angle, due to the projection of the wind along the beam direction (Eberhard et al., 1989). For instance, the observations at 2 km from the lidar (black ring on Figure 3a) are displayed on  Figure 3b and can be fitted by a cosine function in the form of Eq. (1):

$$u_r = a + b\,cos(\theta - \theta_{max}) \tag{1}$$

where $b$ is the mean wind speed, $\theta_{max}$ is the wind direction, $\theta$ is the azimuth angle of the beam and $a$ is the offset (Browning

and Wexler, 1968; Lhermitte, 1962). It is noteworthy that the value of $a$ is much smaller than $b$ for our data. It is possible to retrieve the mean wind from all the "rings" and subsequently calculate the mean wind projected on the beam direction which is displayed on Figure 3c. The difference between the radial wind field $u_r$ (Figure 3a) and the mean wind projected on the beam direction (Figure 3c) is the mlf-cs of the radial wind field $u_r'$ (Figure 3d). A parameter that indicates the existence of a turbulent atmosphere. For this study, the radial wind speed values for which the carrier-to-noise ratio is lower than -27dB (CNR<-27dB)

were disregarded since they were anomalously high, exceeding the values of the rest of the radial wind field by two times or more. Therefore the effective scanning range showcased in Figure 3 is approximately 3 km. For a better visual representation of the patterns, the sign of the $u_r'$ in the current study is positive when the radial wind speed is stronger than the mean wind speed and negative when it is weaker as it is illustrated in the sign convention of Figure 3b and it was computed by the following expression:

$$u_r'= |u_r(\theta)| - |f(\theta)| \tag{2}$$

where f is the fitted curve.

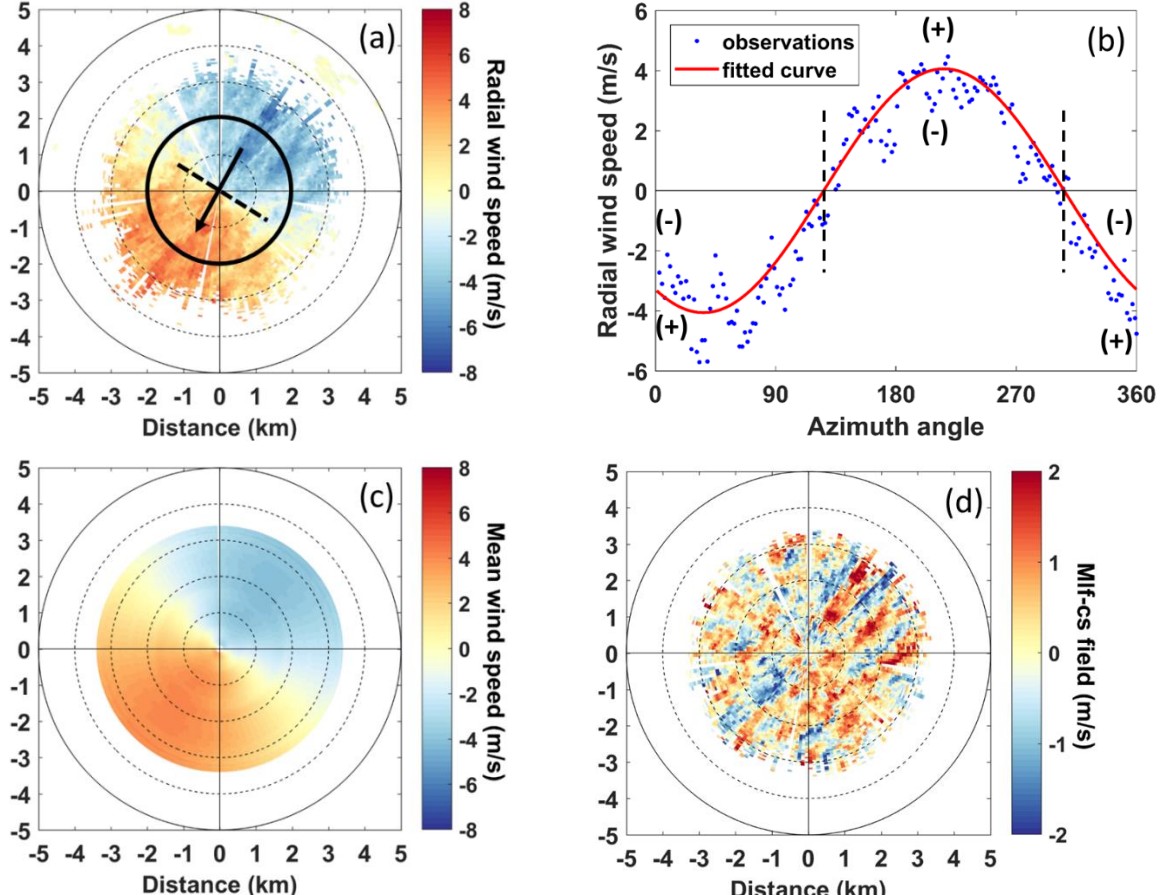

Figure 3: Observations recorded during a quasi-horizontal PPI scan on 08/09/2014 in Jussieu site, Paris at 09:26 UTC. (a) Radial wind speed along with the mean wind direction (black line) and the transverse direction perpendicular to it (black dotted line). (b) Radial wind speed (blue dots) as a function of the azimuth angle at a fixed 2 km distance from the lidar (black circle on panel a) along with the cosine fit function (red line). (c) Mean wind speed projected on the beam direction. (d) Mlf-cs field.

The Jussie site is located in an urban area nearby hills, hence the surface roughness or the orography can affect the regional wind flow. Troude et al. (2002) and Lemonsu and Masson (2002) have performed numerical weather simulations in the area of Paris and have observed that during low wind conditions (below 3 m·s⁻¹) the orographic effect and the urban heat island effect could be the main drivers for the local wind speed. As a result, in some cases the radial wind field does not follow a cosine function, and therefore the VAD method cannot be applied. This is apparent especially at night when low winds

(below 2 m·s⁻¹) do not have a defined direction (Wilson et al., 1976). Figure 4 presents a case where the radial wind field is not homogeneous. The radial wind speed values e.g. at 2 km did not follow a cosine function (Figure 4b).

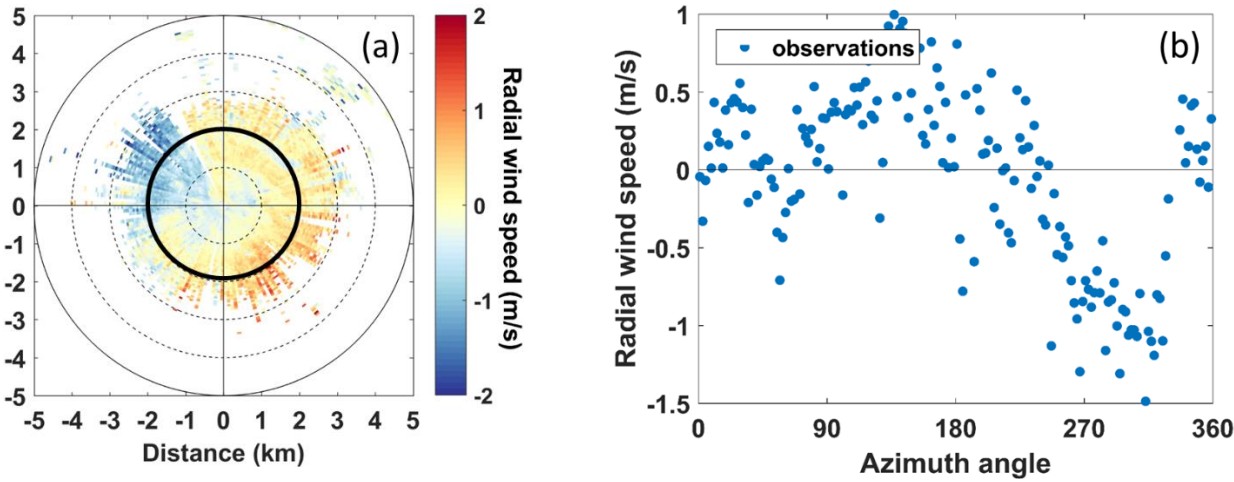

Figure 4: A case when the VAD method cannot be applied: (a) Radial wind field on 25/09/2014 at 23:42 UTC and (b) Radial wind speed (blue dots) as a function of the azimuth angle at a fixed 2 km distance from the lidar (black circle on panel a).

The visual examination of the mlf-cs fields led to the identification of three types of remarkable mlf-cs patterns. The first type was represented by large elongated areas of positive mlf-cs accompanied by large elongated areas of negative mlf-cs aligned with the mean wind (Figure 5a) during the day. In the atmosphere, these types of patterns are encountered concurrently with the existence of rolls, where strong descending motions enhance the horizontal wind speed and ascending motions reduce it. The second type of pattern was characterized by large enclosed areas of positive mlf-cs field attached to large enclosed areas of negative mlf-cs field (Figure 5b) during the day. The convergence zones formed between the positive and negative mlf-cs fields during unstable conditions (e.g. high solar radiation) are able to form strong unaligned thermals. Finally, the third type of pattern consisted of narrow elongated areas alternating between positive mlf-cs and negative aligned with the mean wind (Figure 5c). These patterns resemble turbulent streaks as they are described in Section 1.

In order to train the classification algorithm (Section 4.1), it was necessary to build an ensemble of cases for which the presence of rolls, unaligned thermals or streaks was confirmed by other observations than the lidar measurements. Moderate resolution imaging spectroradiometer (MODIS) true colour images were used to detect the presence of cloud streets over Paris (Figure 5d) which confirmed the existence of rolls as stated in Section 1. Close to the moment when the cloud streets were present, rolls patterns were observed at the turbulent radial fields (Figure 5a). It is noteworthy to mention that, for the training ensemble, we selected only cases of rolls occurring around the satellite overpass time to ensure the presence of cloud streets and thus the existence of rolls. However, for this classification we are interested in all the cases of rolls, with or without the formation of cloud streets. It is important to note that we observed the occurring patterns near the surface, hence near the lower part of the rolls. Regarding unaligned thermals, solar radiation measurements from the meteorological station of Paris-Montsouris indicated the occasions when the hourly values were higher than the monthly average hourly values according to the Photovoltaic Geographical Information System (Huld et al., 2012), signifying fair cumuli weather conditions. For approximately the same time of the day, we observe the unaligned thermals patterns. Figure 5b showcases an example of a turbulent radial wind field with unaligned thermals along with fair weather cumuli over Paris as observed on MODIS true colour image at approximately the same time (Figure 5e).

Finally concerning streaks, a driving factor for their formation is the existence of a strong wind shear near the surface. The observation of the horizontal wind profiles from the DBS scans revealed when the local maxima horizontal wind speed was higher than 2 m·s⁻¹ compared to the local minima above it, which is defined as the threshold for nocturnal low level jet events (Stull, 1988) (Figure 5f). It is important to note that the location of the local maxima and minima of the horizontal wind speed were consistent during the study period ranging from 200 to 300 m and 400 to 500 m respectively. The horizontal wind speed $U_{hor}$ was estimated by the zonal $u$ and meridional $v$ winds via the expression:

$$U_{hor} = \sqrt{u^2 + v^2} \tag{3}$$

For the training ensemble, only night cases when streaks patterns (Figure 5c) were accompanied by differences in local maxima and minima of the $U_{hor}$ higher than 2 m·s⁻¹ were selected. In total, 30 cases of each structure type were selected for the training ensemble with an extra category representing all the patterns that are not classified in the other three categories, such as chaotic patterns or cases when the VAD method cannot be applied (Figure 4). Regarding rolls, streaks and thermals, only cases with symmetric radial wind fields were selected in order to ensure that the VAD method was applicable. The selection of symmetric radial wind fields was based on the visual examinations of the radial wind fields and the individual cosine function fits.

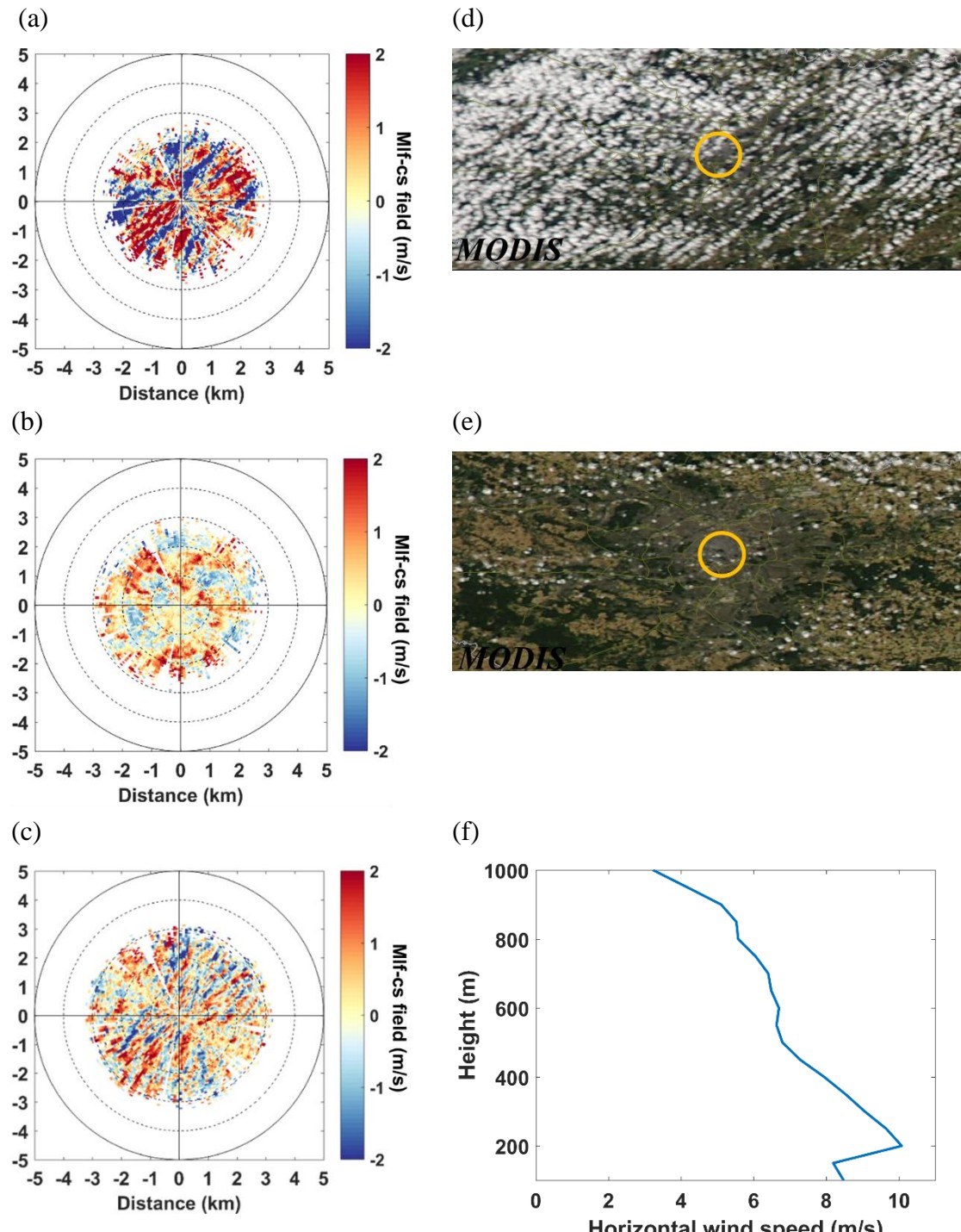

Figure 5: The upper part shows the three types of mlf-cs fields to classify: (a) rolls observed on 13/10/2014 at 12:52 UTC, (b) unaligned thermals observed on 16/09/2014 at 12:52 UTC and (c) streaks observed on 09/09/2014 at 20:49 UTC. The lower part shows the ancillary observations used to ascertain the structure type: (d) and (e) true color image recorded by MODIS Aqua on the same day as (a) and (b) at 12:50 UTC, (f) horizontal wind speed profile recorded by the Doppler lidar using the DBS technique on the same day as (c) at 20:51 UTC.

## 3.2    Computation of the co-occurrence matrices

In order to retrieve comparable texture analysis parameters from the mlf-cs field of the scans, the mlf-cs field was rotated so that the mean wind direction was aligned to the vertical (0° corresponds to a wind blowing from the North). Then, the coordinates were converted from polar to Cartesian. It was also important to adjust the contrast of the image so that the difference between the areas of positive and negative turbulent wind speed became more prominent. For this purpose, the contrast of the images was increased by mapping the turbulent wind speed values into eight levels. One bin included all the

negative values below −0.5 m·s⁻¹, six bins were equally distributed between −0.5 m·s⁻¹ and +0.5 m·s⁻¹ and one bin included all the positive values above +0.5 m·s⁻¹ (Figure 6b).

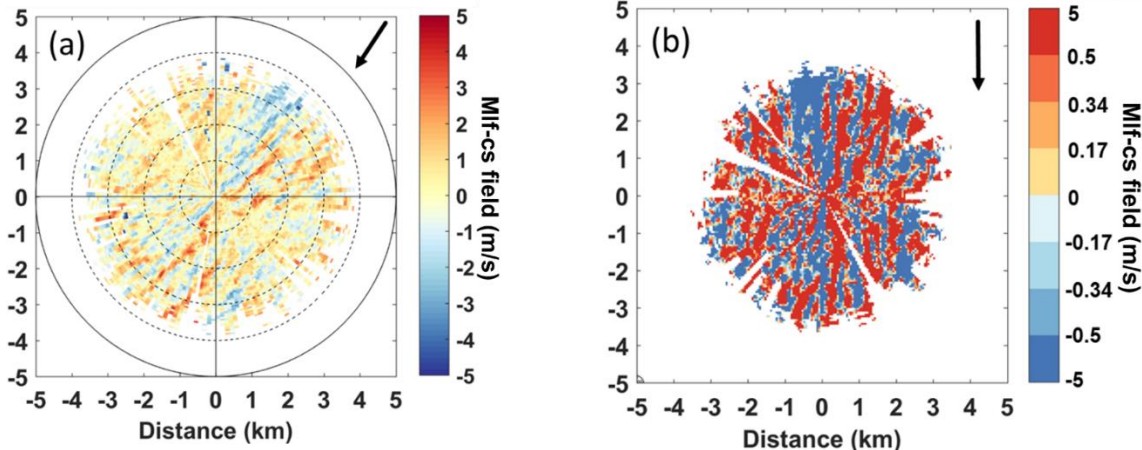

Figure 6: The mlf-cs field (a) before and (b) after image pre-processing with the arrow representing the mean wind direction on 10/09/2014 at 19:57.

For the automated classification of patterns, we need to map them to a space of corresponding numerical parameters. Each reconstructed mlf-cs field is represented by a matrix (cells corresponds to pixels) from which 8×8 co-occurrence matrices (CM) can be constructed (Haralick et al., 1973). The rows and columns of the CM represent the wind levels from 1 to 8, whereas the cells contain the frequency of the combination of two neighbour pixels in the image. More specifically, the element at line $i$ and column $j$ contains the number of pixels with value $i$ which are neighboured by pixels with value $j$. The first neighbour can be searched at different direction (e.g. left-right, up-down or diagonally) defining the cell pair orientation. In the same way a second, a third, etc. neighbour can be selected. Thus, the CM can be calculated for any cell pair orientations and neighbour order. CM were computed for various distances, i.e. neighbour orders $n$ from 1 to 30 (distance from 50 m to 1.5 km) and all possible cell pair orientations, i.e. azimuth angles $\varphi$ from −90° (transverse direction from the mean wind in the counter clockwise direction) to +90° (transverse direction in the clockwise direction). Table 3 shows the cell values of the CM built from the image of Figure 6b for the first neighbour ($n = 1$) and for a cell pair aligned with the mean wind and oriented in the same direction (azimuth $\varphi = 0°$). It is apparent that the vast majority of the occurrences are concentrated in the cells [1,1] and [8,8] as the structures are elongated and aligned with the mean wind direction.

Table 3: Co-occurrence matrix after the image pre-processing (Figure 6b) for the first neighbour ($n = 1$) and for a cell pair aligned with the mean wind and oriented in the same direction (azimuth $\varphi = 0°$).

|   | 1 | 2 | 3 | 4 | 5 | 6 | 7 | 8 |
|---|---|---|---|---|---|---|---|---|
| 1 | 3065 | 226 | 164 | 118 | 113 | 57 | 35 | 94 |
| 2 | 255 | 67 | 77 | 58 | 36 | 26 | 23 | 48 |
| 3 | 181 | 81 | 59 | 61 | 44 | 51 | 35 | 72 |
| 4 | 133 | 58 | 63 | 91 | 71 | 50 | 40 | 92 |
| 5 | 98 | 51 | 59 | 65 | 67 | 63 | 58 | 154 |
| 6 | 58 | 36 | 50 | 53 | 75 | 72 | 78 | 169 |
| 7 | 46 | 30 | 38 | 53 | 60 | 61 | 55 | 231 |
| 8 | 73 | 45 | 78 | 104 | 151 | 201 | 246 | 3402 |

Table 4: Co-occurrence matrix after the image pre-processing (Figure 6b) for the third neighbour ($n = 3$) and for the transverse direction in the clockwise direction (azimuth $\varphi = +90°$).

|   | 1 | 2 | 3 | 4 | 5 | 6 | 7 | 8 |
|---|---|---|---|---|---|---|---|---|
| **1** | 1497 | 231 | 203 | 182 | 165 | 168 | 170 | 1149 |
| **2** | 185 | 19 | 25 | 43 | 27 | 27 | 25 | 200 |
| **3** | 183 | 29 | 26 | 29 | 33 | 31 | 21 | 207 |
| **4** | 195 | 32 | 37 | 39 | 29 | 31 | 28 | 185 |
| **5** | 203 | 29 | 38 | 31 | 36 | 31 | 26 | 208 |
| **6** | 201 | 26 | 25 | 25 | 26 | 39 | 29 | 198 |
| **7** | 175 | 27 | 23 | 26 | 32 | 21 | 37 | 212 |
| **8** | 1063 | 179 | 187 | 196 | 243 | 206 | 217 | 1719 |


On the other hand, Table 4 shows the CM of Figure 6b for the third neighbour ($n = 3$) and for a cell pair oriented perpendicularly to the mean wind (transverse direction) with a clockwise rotation (azimuth angle $\varphi = +90°$). In this case, the occurrences have been distributed to the cells [1,1] and [8,8], as well as to the cells [1,8] and [8,1]. As we can see on Figure 6b, the structures alternate between positive and negative values in the direction transverse to the mean wind, thus creating this

difference in the CM compared to Table 3.

### 3.3    Texture analysis parameters for the classification of the turbulent structures

It is possible to compute several texture analysis parameters from each CM. Srivastava et al. (2018) were able to distinguish different synthetic patterns by using four texture analysis parameters: correlation, contrast, homogeneity and energy. Correlation indicates the existence of linear structures in the image, with high values associated to a large amount of

linear structure in the image. Contrast reveals the local variations in an image, where a large amount of variations leads to high values. Homogeneity is self-explanatory and the high values represent a homogeneous image. Finally, energy measures the uniformity of an image with the highest values corresponding to constant or periodic forms (Haralick et al., 1973; Yang et al., 2012). In their study, the striped patterns resemble the elongated patterns of streaks and rolls that we observe in the radial turbulent wind field. Therefore, the same texture analysis parameters were selected for calculation in our dataset. More

particularly, these parameters were computed by the Eq. (4), (5), (6) and (7):

Homogeneity:

$$Hom(\varphi, n) = \sum_{i,j} \frac{p(i,j)}{1 + |i - j|} \tag{4}$$

Contrast:

$$Con(\varphi, n) = \sum_{i,j} p(i,j)|i - j|^2 \tag{5}$$

Correlation:

$$Cor(\varphi, n) = \sum_{i,j} \frac{(i - \mu_i)(j - \mu_j)p(i,j)}{\sigma_i \sigma_j} \tag{6}$$

Energy:

$$En(\varphi, n) = \sum_{i,j} p(i,j)^2 \tag{7}$$

where $p(i,j) = \frac{CM(i,j)}{\sum_{i,j} CM(i,j)}$ for the $i, j$ position in the CM, marginal expectations

$\mu_i = \sum_i \sum_j i \cdot p(i,j)$, $\mu_j = \sum_i \sum_j j \cdot p(i,j)$ and the marginal standard deviations

$\sigma_i = \sqrt{\sum_i \sum_j (i - \mu_i)^2 \cdot p(i,j)}$, $\sigma_j = \sqrt{\sum_i \sum_j (j - \mu_j)^2 \cdot p(i,j)}$.

At a given neighbour order $n$, it is then possible to study the dependence of the texture parameters to the azimuth angle $\varphi$ (see an example of such a dependence on Figure 7). The streaks and rolls have a more prominent peak in the longitudinal direction ($\varphi = 0°$) compared to the unaligned thermals and "others" patterns. As streaks and rolls are aligned with the mean

wind (azimuth $\varphi = 0°$), those peaks result from the elongated shapes of these patterns.

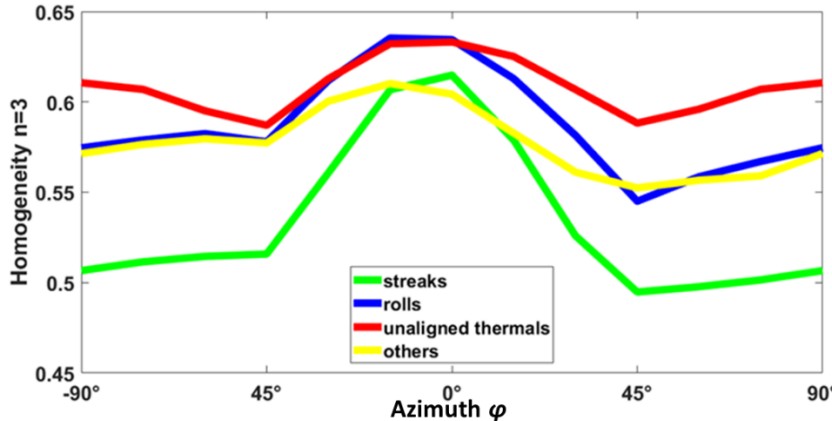

Figure 7: Third neighbour homogeneity as a function of azimuth for one selected scan of each type.

Three parameters of the curve in Figure 7 were selected in order to distinguish the different types of structures. For instance, for the homogeneity curves, these parameters are defined by the Eq. (8), (9) and (10):

Amplitude:     $Hom.\,Amp(n) = max_\varphi\big(Hom(\varphi, n)\big) - min_\varphi\big(Hom(\varphi, n)\big)$                        (8)


Integral:         $Hom.\,Int(n) = \sum_\varphi Hom(\varphi, n)$                                               (9)

Symmetry:     $Hom.\,Sym(n) = \sum_\varphi |Hom(\varphi, n) - Hom(-\varphi, n)|$                          (10)

These three curve parameters were calculated for the four texture analysis parameters and for each of the thirty neighbour orders, which gives 360 parameters. In addition to these parameters, the UTC hour (close to solar time in Paris), the average mean wind speed and the root-mean-square error of the cosine fit (Figure 3b) were included in the classification

parameters. The total number of classification parameters associated with each scan was therefore 363.

## 4.    Classification using supervised machine learning

### 4.1    Algorithm training and classification error

In order to classify the mlf-cs according to the aforementioned texture analysis parameters, the supervised machine learning methodology was applied (Bonamente, 2017; James et al., 2000; Kubat, 2017). The QDA algorithm was used, that

minimizes the total error probability of the classification, assuming that features of each class have a multidimensional Gaussian distribution. QDA or normal Bayesian classification (Hastie et al., 2009) is the parametric approach implying that probability density functions (PDF) belong to the family of normal distributions. It is a classical algorithm of the supervised

machine learning, based on the principle of maximum likelihood. The general idea is to estimate the PDF for each class, and then select the most probable class (Kubat, 2017).

The greedy algorithm of stepwise forward selection was used in the article, which is the standard and frequently used method of reduction of the feature space. As indicated in (Sokolov et al., 2020), it can be formulated as follows. The features are divided into two groups: accepted in the classification model and remaining, for which an estimate of the possibility of acceptance into the model is checked. Features from the set of ''remains'' are consecutively added to the model and corresponding estimations of the classification error are calculated. From the received set of errors, the minimum is chosen

and compared with the error of the previous model. If a significant reduction of the error occurred, then the corresponding feature is accepted into the model, if not then the process stops. The QDA was trained (Hastie et al., 2009; Sokolov et al., 2020) with the 150-case ensemble described in Section 3.1: 30 cases of streaks, 30 cases of rolls, 30 cases of unaligned thermals and 60 cases of "others". The category of "others" was represented by twice more cases since it is expected to be the dominant category in the classification, as it includes the chaotic mlf-cs fields and the cases where the mlf-cs field was not computed

successfully by the VAD method. The algorithm can be sensitive to an unbalanced training ensemble. Therefore, the selection of a training ensemble based on the expected results was preferred (Kubat, 2017).

        The classification error of the QDA technique could be estimated for the training ensemble by means of the 10-fold cross validation. In this method, the algorithm is trained using 90 % of the training ensemble (135 cases), then it is applied to the remaining 10 % (15 cases) and the resulting (output) classes are compared to the expected (target) classes. The process is

repeated 10 times, each time extracting a different 10 % sample for test, until the entire training ensemble has been tested.

        As the number of dimensions of the feature space (363) was significantly higher than the number of patterns of the training ensemble (150), the application of all the features leads to the curse of the dimensionality problem, when the classification works well only for the training data and fails for the test set. In order to deal with this problem, we reduced the feature space by selecting the most informative components using the stepwise forward selection algorithm (Sokolov et al.,

2020). The resulting sequence of these components and the decrease of the 10-fold cross validation classification error are presented in Figure 8. The classification error reached a minimum of about 9.2 % when five parameters were used; taking more into account increased the classification error.

        Analytically, these parameters are the amplitude of the 2nd-neighbour homogeneity curve, the integral of the 18th-neihgbour contrast curve, the amplitude of the 4th-neighbour contrast curve, the integral of the 8th-neighbour correlation curve

and the symmetry of the 2nd-neigbour homogeneity curve. These results show that the prominent peaks are a distinctive characteristic for the elongated patterns as the amplitude of the homogeneity and contrast curves are two of the significant parameters. Furthermore, the integral or more precisely the sum of the points of the curves for the contrast and for the correlation curves are significant parameters as well. This is important especially for the distinction between the categories thermals and "others" as their amplitude may not differ substantially since the patterns are not towards a specific direction, yet

a chaotic area will have higher values of contrast and lower values of correlation compared to an enclosed homogeneous area. Finally, the symmetry of the homogeneity curve as a classifier reveal the urgency to align the radial turbulent wind fields to the mean wind direction and thus align the structures such as streaks and rolls with the mean wind direction in order to be distinguishable from the random positions of the enclosed structures of the thermals or the chaotic structures of the "others". It is also crucial to note that the parameters cover various distances, from the 2nd-neighbour, which in grid points is 100 m to

the 18th-neighbour which is 900 m. This is necessary for our classification since streaks and rolls are both elongated patterns but their transverse horizontal sizes differ. Furthermore, it demonstrates the ability of the algorithm to distinguish structures with different sizes. It is noteworthy that the curve parameters play a more significant role in the classification of the structures in comparison to time, mean wind field and cosine fit RMSE.

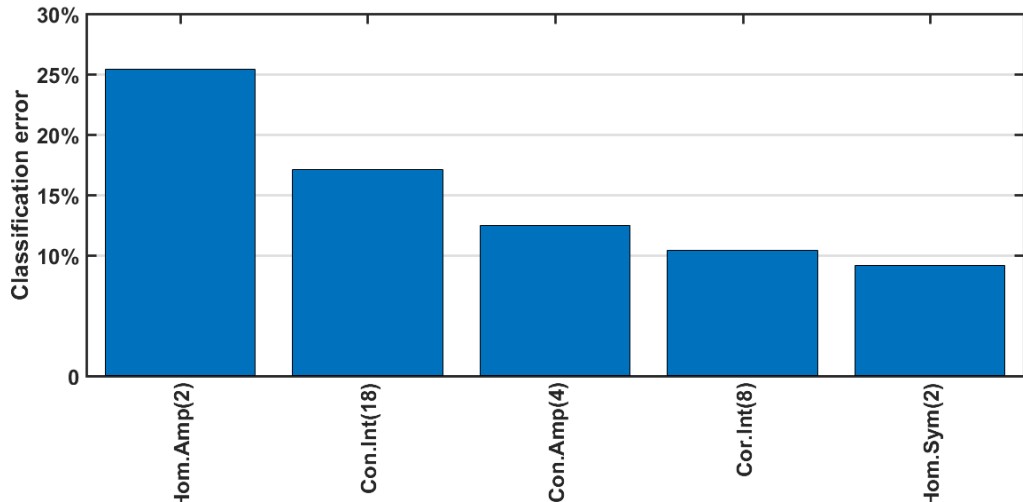

Figure 8: Parameters selected to minimize the classification error of the training ensemble by the QDA method. From left to right: Amplitude of the homogeneity for the 2nd-neighbour, integral of the contrast for the 18th-neighbour, amplitude of the contrast for the 4th-neighbour, integral of the correlation of the 8th-neighbour and symmetry of the homogeneity for the 2ndneighbour.

The detailed results of the cross-validation of the QDA classification for the algorithm with five predictors are displayed in Table 5. The algorithm allowed classifying correctly about 91 % of the training ensemble. The algorithm performs the most precise classification for the streaks with a classification error of only 3.3 % as one case was misclassified as rolls. Regarding the category "others", the results are equivalently accurate with a classification error of 3.3 % as two cases were misclassified as thermals. Moreover, the performance of the algorithm for rolls was good with a classification error of 10 % with 3 cases were misclassified as thermals. Thermals were the most troublesome type for classification by the algorithm, the algorithm classified correctly 24 cases. Four cases were misclassified as rolls and 2 cases as "others" showing a classification error of 20 %.

Table 5: Confusion matrix calculated for the training dataset. The "target class" corresponds to the visual classification while the "output class" corresponds to the class attributed by the algorithm. Therefore, the cells in the "roll" column, for instance, give the number of roll cases that were classified properly (roll line) or improperly (other lines) in the different categories.

| Target class / Output class | Others | Streaks | Rolls | Thermals | |
|---|---|---|---|---|---|
| **Others** | 58 | 0 | 0 | 2 | 96.7 % |
| | 38.7 % | 0.0 % | 0.0 % | 1.3 % | 3.3 % |
| **Streaks** | 0 | 29 | 0 | 0 | 100.0 % |
| | 0.0 % | 19.3 % | 0.0 % | 0.0 % | 0.0 % |
| **Rolls** | 0 | 1 | 27 | 4 | 84.4 % |
| | 0.0 % | 0.7 % | 18.0 % | 2.7 % | 15.6 % |
| **Thermals** | 2 | 0 | 3 | 24 | 82.8 % |
| | 1.3 % | 0.0 % | 2.0 % | 16.0 % | 17.2 % |
| | 96.7 % | 96.7 % | 90.0 % | 80.0 % | 92.0 % |
| | 3.3 % | 3.3 % | 10.0 % | 20.0 % | 8.0 % |

## 4.2 Results of the trained algorithm over the 2-month dataset

The whole dataset, consisting of 4577 scans, was classified according to the five parameters showcased in Figure 8. The results are displayed in Figure 9.

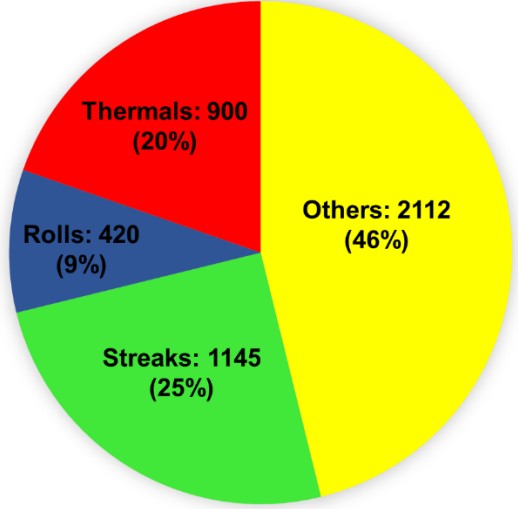


Figure 9: Classification of the whole ensemble using the QDA method according to the parameters of Figure 8.

The algorithm classifies 54% of the two-month dataset as containing mlf-cs and 34 % in particular as coherent structures (streaks, rolls). The most frequent cases of mlf-cs were streaks (25 %) and the least frequent were rolls (9 %). It is important to note that, in our classification, we considered only thermals and rolls during daytime. Figure 10 illustrates the number of
occurrences for each type of structure at a particular time of the day during the two months of the campaign. It is evident that despite time was not one of the selected classifiers, the number of occurrences of the structures show a distribution that can be associated to the atmospheric conditions. More particularly, rolls and thermals were mainly classified during day. This result is noteworthy as these structures are linked to a well-developed atmospheric boundary layer during day. On the contrary, there were scarcely any rolls cases observed at night and a few unaligned thermals were classified at night. This stems from the
training process, where some cases of thermal were improperly classified as "other" and the reverse. Regarding the "others" cases, these were mostly observed during the night. This was expected since the cases of low winds with no defined direction –when the VAD method cannot be applied– occur mainly during the night. We also see that streaks were observed more frequently during the night, when mechanical turbulence becomes dominant. This was also expected as the nocturnal low level jets is a main driving factor for the formation of streaks and we observed the occurrence of the local maxima of the horizontal
wind speed near the surface higher than 2 m·s$^{-1}$ compared to the local minima over Paris for 20 out of the 62 nights during the VEGILOT campaign.

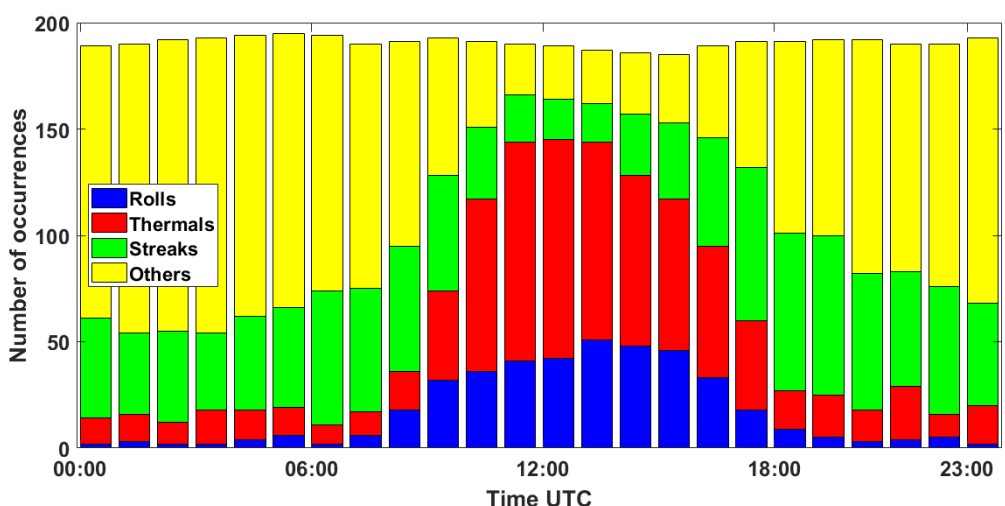

Figure 10: Histogram of the number of occurrences of the different types of structures with respect to time in UTC.

## 5. Conclusions

The current study showcases that it is possible to identify and classify mlf-cs such as streaks, rolls and unaligned thermals with horizontal scans from a single Doppler lidar by combining texture analysis parameters and the QDA supervised machine learning technique. By applying the VAD method to the radial wind observations, it is possible to identify mlf-cs that can be distinguished to narrow elongated (streaks), wide elongated (rolls), large enclosed (thermals) and chaotic ("others") patterns. These diversities of the patterns were also depicted in the curves of the texture analysis parameters with the elongated patterns (streaks, rolls) showing a prominent peak compared to more chaotic or enclosed patterns (unaligned thermals).

A training ensemble of 150 cases was selected by combining visual examination of the patterns and studying characteristic physical properties corresponding to streaks, rolls and unaligned thermals. Subsequently, the QDA algorithm with stepwise forward selection of the features was applied to the training ensemble and its' performance was estimated using the cross-validation technique. The results showed a successful classification for 91 % of the training ensemble using five texture analysis parameters as predictors. More particularly, these parameters were the amplitude of the $2^{nd}$-neighbour homogeneity curve and the amplitude of the $4^{th}$-neighbour contrast curve which were associated to the prominent peaks of the elongated patterns (streaks, rolls). Furthermore, the integral of the $18^{th}$-neihgbour contrast curve and the integral of the $8^{th}$-neighbour correlation curve which could distinguish, for example, chaotic patterns ("others") with high contrast and lower values of correlation between neighbour points compared to an enclosed homogeneous (thermals). Finally, the symmetry of the $2^{nd}$-neigbour homogeneity curve revealed the importance to align the mlf-cs fields to the mean wind direction. Another striking outcome of the QDA classification was the variety of the classifiers in terms of distance between the grid points. The $2^{nd}$-neighbour translates in a distance between two grid points equivalent to 100 m and for the $18^{th}$-neighbour to 900 m. This is essential for the classification between patterns with different sizes such as streaks and rolls. The algorithm performed best for the category of streaks with a classification error of only 3.3 %. Time, mean wind speed and the cosine fit RMSE of the VAD method were not selected by the algorithm for the classification.

The whole ensemble of the 4577 scans was classified by the trained QDA algorithm using the five selected texture analysis parameters. The results showed that 54 % of cases were classified as mlf-cs among which 34 % were coherent structures (streaks, rolls). The streaks were mostly observed during night whereas the thermals and rolls were almost exclusively observed during the day, with only a few cases classified between sunset and sunrise. The classified ensemble can be used for statistical studies of the mlf-cs physical parameters, such as structure size as a function of weather conditions (PBL height, temperature, wind speed, radiation etc.). Moreover, the development of the structures can be analysed and comprehended.

### Data availability

All lidar data used in the study are property of the Laboratoire de Physico-Chimie de l'Atmosphère (LPCA), Dunkirk, France and are not publicly available. MODIS satellite images are publicly available following NASAs' open data policy (https://earthdata.nasa.gov/collaborate/open-data-services-and-software).

### Author contribution

IC, EDE, HD and AS conceptualized this study and developed the methodology. HD, PA and MF installed and monitored the instrument on the field. IC processed the data and analysed the results for all parts of the study, with the help of HD, AS and EDM for Section 4. IC wrote the original draft of the manuscript, with contributions from HD, EDE and AS. All authors participated in the review and editing of the manuscript and agreed to this version.

**Competing interests**

The authors declare that they have no conflict of interest.

**Acknowledgements**

The authors thank F. Ravetta, J. Pelon, G. Plattner and A. Klein of the LATMOS, Sorbonne University, Paris for organizing and carrying out the VEGILOT campaign.

This work is a contribution to the CPER research project IRenE and CLIMIBIO. The authors thank the French "Ministère de l'Enseignement Supérieur et de la Recherche", the "Hauts-de-France" Region and the European Funds for Regional Economic Development for their financial support to this project. The work is supported by the CaPPA project. The

CaPPA project (Chemical and Physical Properties of the Atmosphere) is funded by the French National Research Agency (ANR) through the PIA (Programme d'Investissement d'Avenir) under contract "ANR-11-LABX-0005-01" and by the Regional Council " Nord-Pas de Calais » and the  "European Funds for Regional Economic Development (FEDER).

We acknowledge the use of imagery provided by services from NASA's Global Imagery Browse Services (GIBS), part of NASA's Earth Observing System Data and Information System (EOSDIS).

Experiments presented in this paper were carried out using the CALCULCO computing platform, supported by SCoSI/ULCO (Service COmmun du Système d'Information de l'Université du Littoral Côte d'Opale).

This study was funded by RFBR, project number 20-07-00370.

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
