# Peer review of "Detecting turbulent structures on single Doppler lidar large datasets: an automated classification method for horizontal scans"

_Atmospheric Measurement Techniques, 2020_

## Referee Comment (RC1) · Anonymous Referee #1 · 21 Jun 2020

General Comment:

This manuscript is presenting a method to identify coherent turbulent structures using scanning wind lidar data. The method is based on the application of a machine learning algorithm to detect different patterns of spatial variations of the wind speed. The usage of this methodology to categorise data acquired by a scanning wind lidar instrument is innovative and of interest for the remote sensing research community. Its application can contribute to the analysis of large data sets by minimising the necessary amount of processing time that is required today to categorise scanning wind lidar data.

1. In the manuscript a cosine fit is applied to wind lidar data acquired during VAD scans,

in order to estimate the mean wind speed and direction. Such a data analysis method requires the assumption of horizontal homogeneity of the wind vector, as also stated by the authors. However, I am wondering to which extent this homogeneity is expected over the urban landscape of Paris. I think that a discussion on the terrain heterogeneity is missing from the article. In this direction, I think that it would be constructive to add an elevation map of the area over which the scanning wind lidar was acquiring measurements. This would provide an insight to which extent the observed spatial variations of the wind are related to temporal fluctuations of the wind and/or due to changes of the terrain elevation.

In this context and regarding the data analysis:

a. A statistical parameter is required to specify the representativeness of the fit used in the VAD scan. In the manuscript it is stated that the RMSE values of the estimated fit have been calculated but they are not stated in the document.

b. Did the authors perform a quality check of the acquired data? Do they apply any SNR filtering to the acquired radial wind speed prior the application of the data analysis?

2. The subtraction of the mean wind speed from the radial wind speeds does not compensate the fact that the individual measurements along the scanning pattern are the result of the projection of the instantaneous wind vector to the line-of-sight of the wind lidar. Therefore, the term turbulent wind field could lead to a misinterpretation. The authors should clearly state that they measure the high frequency fluctuations of the radial wind speed.

Specific Comments:

P2 Line 28. The reference of Roth 2007 is a review of the atmospheric turbulence studies over urban landscapes and it doesn't directly discuss how the pollution concentration in urban environment is dependent on the weather and on the turbulence. A reformulation of this sentence is suggested for a clarification.

P3 Line74. Why is it relevant if the campaigns are short-term or long-term? And what is the time scale of these two types of campaigns that it is relevant for the topic of this study?

P3 Line95. I am not sure that the two references of Kumer 2014 and Veselovskii et al. 2016 are the appropriate to be used here. In none of them there is an analytical explanation of the wind lidar instrument as it is stated. I suggest replacing them with more relevant references. An example could be the work of:

Cariou, J. P., R. Parmentier, M. Valla, L. Sauvage, I. Antoniou, and M. Courtney. "An innovative and autonomous 1.5 $\mu$m Coherent lidar for PBL wind profiling." In Proceedings of 14th Coherent Laser Radar Conference. 2007.

P3 Line 101. It is stated that "The duration of each scan was 3 minutes which is sufficiently short for the observation of structures". Could the authors elaborate more on this statement?

P4 Line 102. The authors state that the maximum range of the scans reached 5 km. However, in none of the figures that they have included in the manuscript the range ever reaches 5 km. The most common range in their data is between 2 – 2.5 km. Can the authors explain why is this happening?

P4 Line111. Table 2. A list of scanning patterns is included here that are not used in this study. In addition, the purpose of each scanning pattern is included without explaining the reasoning for their selection. I would suggest to either remove this table. If the authors wish to keep it then I suggest that they should elaborate more in the text about it.

P4 Line120. The "offset" refers to the vertical wind speed component? What does it mean that $\alpha$ is much smaller than b? Is this a common observation over the whole scanning plan? And what kind of $\alpha$ values does the application of the model result in?

P6. Line156-157. It is not clear how the unaligned thermals are dependent on the

increased solar radiation measurements.

P6. Line 163. How do the authors estimate the value of the wind shear? Also, shouldn't the units of the wind shear be seconds to the power of -1?

P6. Line 166. The authors in P5. Line 133 state that the VAD method could not be applied in the data acquired during the night, especially at those occasions where the mean wind speed was less than 2 m/s. Does this mean that only cases with mean wind speed higher than 2 m/s were selected?

P6. Line 167. How well was the mean direction and speed estimated through the VAD method in the selected cases? I suggest adding a statistical parameter that describes the representativeness of the applied fit (equation 1) to the measured data (e.g. RMSE).

P7. Line 170. It is difficult to understand the scale of the Modis colour images. Would it be possible to either add a scale or mark on the images the scanning area?

P7. Line 176. I recommend describing very shortly the texture analysis especially in the context of remote sensing. It would be useful to add any references in the introduction regarding the previous applications of this type of analysis to remote sensing data.

P8. Line 181-182. What is the logic behind the selection of this particular values for defining the contrast?

P10. Line 243. The authors state that 60 cases of "other" patterns are used during the supervised machine learning step. They argue that this is necessary because "it is expected to be the dominant category in the classification". I am not sure that I understand what it is meant with this statement. I suggest having this part a bit more explained. Furthermore, how is the mean wind speed and direction estimated in these cases? As it is stated the VAD was not successfully applied to this data.

P10. Line 225. How do the authors explain the change in the slope of the homogeneity curve that is observed for absolute azimuth angles larger than 45 degrees?

P10. Line 240. Could the authors add a reference to the literature describing the “supervised machine learning methodology”?

P11. Line 265. How do the authors physically explain this result? A low RMSE in the cosine fit, couldn’t also mean that the mean wind speed and direction are not estimated correctly?

P12. Line 291. It is stated that “there were scarcely any rolls cases observed at night”. However, in the sentence 288 it is written that for the classification of this study the authors “consider only thermals and rolls during daytime”. To my understanding there is an inconsistency between the two statements. Can this be explained better?

P13. Line 312. In the conclusions the author state that time, wind speed and the cosine fit RMSE of the VAD method were not selected by the algorithm for the classification. However, in the results presented in figure 9 there is a time dependency in the detection of certain patterns (e.g. thermals and rolls). Could the authors comment why the inclusion of the time as a classification parameter would not improve further their results?

P13. Line 318. Given the fact that one PPI scan lasts for 3 minutes and occurs every 18 means, can the authors explain how does the acquired data set contribute to the comprehension of the development of coherent structures?

Technical Corrections:

General: There is a small inconsistency on the way that figures are referred to. Sometimes they are used parenthesis after the number of the Figure to denote a subfigure and sometimes are not.

P2 Line 34. I think that the statement “Futhermore, ....” is a self-evident. I would suggest removing it. Also, I would suggest to add the reference of (Hussain, 1983) at the end of the previous sentence.

P2 Line 37. “and the lower”

P3 Line 72. I suggest reformulating this sentence. It is not clear to the reader how is the two-dimensional autocorrelation function was used. Also, I suggest changing the sentence “the observation of the scans by eye” to “visual observation of the scans”.

P5 Line132. The authors state that due to the surface heterogeneity the VAD method can be applied in some cases. A surface heterogeneity will introduce an error in the VAD method regardless the wind speed.

P3 Line76. Change “month” to “months”

P3 Line83. The “Section 0” should change to “Section 3”

P3 Line86. The word “two” is used as a noun modifier and therefore the word month should be in singular form.

P3 Line96. I suggest to change the text “The lidar is sensitive only to the” to “A wind lidar is measuring the”

P3 Line100. Change the “for a” to “with a”.

P4 Line103. I think that it is more grammatically correct to either use the past tense or the passive form of the “rise” verb.

P5 Line132. I suggest that the “Jussie site” is changed to “The Jussie site”.

P5 Line138. Figure 3 caption. Change the “a case” to “A case”, also a add a tab space between (a) and “Radial”.

P6. Line149. “Sec” should be replaced by “Section”.

P6. Line 164. I suggest to re-write the sentence “For many cases, the wind shear was accompanied by turbulent streaks pattern” and specify for which particular wind shear values were streaks detected.

P6. Line165. The part of the sentence “so or the training ensemble” should be rewritten.

P10. Line 230. “fore” should be changed to “for”.

P11. Line251 – 253. The authors state the higher number of dimensions relative to the number of patterns lead to the “curse of the dimensionality problem”. I suggest to re-write it by using appropriate scientific statements.

P11. Line 259. Correct the “Section 0”.

P12. Table 5 caption: I suggest changing the “eye-made” to “visual”.

P12. Line 290. The coherent structures don’t have a preference. They are formed under favourable atmospheric conditions. I suggest commenting the result of Figure 9 on that basis.

P15. Line 386. “Lemone” should be changed to “LeMone”. Also, necessary information of the article (e.g. journal) is missing.

---

## Referee Comment (RC2) · Anonymous Referee #2 · 23 Jun 2020

**1 General comments**

The authors present the implementation of an innovative approach for identification and classification of coherent structures in atmospheric flow over urban canopy. The approach is based on measurements from a single Doppler lidar system installed within the surface layer, covering a significantly large spatial domain, and uses a machine learning technique to classify spatial patterns in horizontal wind fields under different stability conditions. The approach shows advantages over classic methodologies for coherent structure identification. It reduces data processing time, and it might allow a more automatic structure detection from large databases.

As I understand form the first paragraph in the introduction, comprehension of the flow physics it is important for monitoring atmospheric pollution. However, the physics identified here, in the form of coherent structures, are not related with bad pollution conditions, since the latter might fall in the “Other” category, and their physics are not clear from the study. The motivation of the study should be stressed form the beginning. Only when we arrive to the conclusions we can read some of the potential application of this approach. (line 316 of the text).

There are many previous studies on coherent structures, as well as lidar technology, than could be included. The description of the coherent structures that this study aims to identify is a bit vague and needs improvement.

In general, the term turbulence and turbulence fields are used frequently in the text, but the range gate resolution of the lidar scanner is 50m at a height of 75m above ground level. Turbulence and its most energetic eddies might fall within this length scale. Almost all turbulence fluctuations are filtered out by the lidar due to spatial averaging. What we can clearly see from lidar observations is medium-to-large scale fluctuations and coherent structures rather than turbulence.

The methodology behind data process could be better explained. I miss a paragraph describing how data quality was assured. Did you use a CNR/SNR threshold for filtering? What was the data recovery rate during the two months of measurements?

It is not clear from the text how the radial wind speed fluctuations are calculated. What does “stronger radial wind speed” mean? A larger absolute value?. It seems to me that the sign might come from the combination of $u$ and $\cos(\theta)$. One suggestion is to put this definition as equations. Since the wind direction is obtained from equation (1) it would be possible to work with the streamwise component $u$ instead of the radial wind speed $u_r$.

It is not clear from the results, what is the relative importance of non-distinguishable structures, bad fittings, and bad data (low CNR/SNR signals) in the “Others” category.

The authors give some information about what it seems to be the reason of one group of cases to belong to this category (bad fitting of cosine function), but no threshold on this fitting error is given.

I miss more elaboration in the description of the texture parameters used for classification, namely, why they might be relevant and if they were relevant in the end. The feature selection process is also not very clear. Cross validation is well known and well explained, but the text explaining the outcome as well as the figure used in that regard are confusing.

A brief description of the machine learning technique used could be useful for clarity. Conclusions section. In my opinion, this section should be read in a positive rather than negative way. Example: it should focus on the relevant parameters discovered (which need a bit more explanation in the corresponding section) rather than the ones excluded by the study. The sections describing the methodology used-which need some improvement-are already clear, and no repetition is needed. Same with the results highlighted.

**2 Specific comments**

Page 1 Line 12. Change ”manually” to “visually”.

Page 1 Line 15. Change “and installed” to “installed”.

Page 1 Line 16. It would be better to reword this sentence, maybe “The turbulent component of radial wind speed is estimated using. . .over 4577 scans.

Page 1 Line 18-21. I am not sure what the sentence describing the training set adds to the abstract if not combined with the next one. It is better to state directly the unsupervised algorithm used instead of using parenthesis. It might be better to rephrase this in a more concise way.

Page 1 Line 23. What are the remaining 20

Page 2 Line 26. Change “step for” to “step towards”.

Page 2 Line 32. A coherent structure is defined according to its phase-averaged rather than its instantaneous vorticity. I also suggest moving the Hussain 1983 reference to this sentence. A coherent structure needs to maintain its phase-averaged vorticity rather than its time-averaged vorticity or form.

Page 2 Line 35. Please specify that this is the case of atmospheric flow. Other structures are observed at laboratory scale (also in the atmosphere but not so relevant for momentum or scalar fluxes), like hairpins, or hairpin trains. Include a reference to Hutchins and Marusic (2006) and Adrian (2007).

Page 2 lines 37-44. Consider reordering the sentences here for a more fluent reading. Maybe starting the paragraph from sentence in line 41?.

Page 2 line 45. How is it that you identify rolls in the mixed layer, with sizes from few to dozen kilometers, with scans at surface layer height (75m) with spatial coverage of less than 2 kilometers?. Is this description coherent with what you are identifying?.

Page 2 lines 57-65. Since you are using lidar instead of radars it would be better to shorten the scanning pattern description using some of the given references, since they have to do only with the history on scanning patterns. There are more recent references of this regarding lidars. Cariou (2007) and Vasiljevic (2016).

Page 3 line 72. Replace “by eye” by “visual inspection” or similar.

Page 3 line 73. More than time-consuming, it might be non-systematic.

Page 3 line 74. You meant “A less time-consuming” approach? What height was the met mast?

Page 3 line 78. “This study aims to identify turbulent coherent structures from single Doppler lidar horizontal scans”. Also, please introduce here what is texture analysis

(roughly maybe) and what what machine learning technique you are using.

Page 3 line 83. Section 0 must read section 3, here and in the rest of the text.

Page 3 line 86-91. It should read “measurement campaign”. Move “in Paris” to the end of the sentence, modified to “in the urban area of Paris”. Remove the url of leosphere to the reference section maybe. More than only sensitive to the radial component, the lidar does measure and it is intended to measure only the radial component. Lidars technology and its operation principle is well known, use references (Cariou, maybe write ts paragraph in amore concise way, being specific in the corresponding table than in the text here, which is a bit confusing.

Page 3 line 101. I would say fast instead of short. Also, what type of structures and why this time window?

Page 4 line 121. What is the reason behind a is small for your case?.

Page 5 Figure 2. Why the reach of the lidar is 2 km and no 5 km?.

Page 5 line 134. Is it possible to specify the fraction of cases with low winds, and its relative importance to the number of bad fittings?. I miss an analysis of stability conditions, since it seems that stable conditions affect the most.

Page 6 line 143. Actually, for rolls, it is the opposite. Ascending motions bring low momentum to higher levels, reducing the speed, and vice versa.

Page 6 line 146. Since rolls and streaks both present areas of alternating low/high momentum with elongated shape, their main difference is their extent. What is the criteria to differentiate between them? The clouds formation shape from MODIS was used, as I understand, only for a fraction of the cases included in the training dataset.

Page 6 line 161-165. Wind shear is defined as du/dz with 1/s units, could you clarify what definition you are using here? Additionally, streaks are present in turbulent flow as well, beyond stable conditions, why do you focus in cases with low turbulence energy

(stable conditions)? It seems that high shear due to jets is only one among several mechanisms.

Page 6 line 166. How many cases did you use for the “Other” category? From table 5 seems that they are around 60, the double. What is the reason for such big number?. Can this influence the final classification output? This is explained in section 4, but it should be clear from here.

Page 7 Figure 4. What is the scale of map in (d) and (e)?

Page 7 line 176. Could you introduce what texture analysis is first? Additionally, since “Others” had a poor fitting and then uncertain wind direction, how did you align them with 0 degrees?

Page 7 line 180. Eight bins were chosen for increased contrast. Why eight?, could you develop more on this?. What is the effect of the number of bins in the output?

Page 8 line 185. The procedure for the construction of the CM matrix is a bit confusing. Could you write it in a more concise way?.

Page 9 line 212. Is it possible to elaborate more on the 4 parameters described?. It is not clear only from the equations what their characteristics are.

Page 10 Figure 6. The notation of the azimuth angle is different form the text. Why does homogeneity grow after 45 degrees for all categories? The definition of homogeneity says that CMs with large values in the diagonal might result in larger values of this parameter. The diagonal from table 3 to 4 decrease because of azimuth angle. Should homogeneity decrease monotonically from 0 to +/- 90 degrees?. Can you elaborate more on this?. How many cases are represented for each category in the figure? Only one scan? An average from many cases?

Page 10 line 231. Notation is a bit weird here.

Page 10 line 241. The description of the training set might be better place in section

3. Why is it expected that “other” category should double the rest? Please elaborate more on this.

Page 11 Figure 7. This figure is very confusing and not self-explanatory at all. Please give more information in its caption, relative to the number in parenthesis (neighbor order I suppose), state that they are all or a few of the final parameters used.

Page 12 Table 5. Change “eye-made” to a better term, like visual classification or similar.

Page 12 line 286. It is not clear if streaks were also detected during daytime, since the previous definition of the training set (line 162) says only night-time, but figure 9 says the opposite. Same for rolls and thermals. In summary, the constraint you talk about (day-time rolls and thermals, night-time streaks) does concern only the training set definition?.

Page 12 line 292. If I am correct, you tried to explain thermals during night, not others during days. Only the last word in the sentence, “reverse”, explains this. Moreover, can you elaborate more on what is the reason behind the erroneous classification of thermal as “others”? During stable conditions turbulent eddies are smaller, structures also show smaller length-scales. However, mean wind can show slight differences with no directional preference, and they can look like thermals (see Shah and Bou-Zeid, 2014).

Page 13 line 295. Stable cases during night show buoyant forces opposing vertical momentum flux and turbulence generation. Mechanical turbulence does die out under stable conditions. Mechanical turbulence destruction by buoyancy is the dominant mechanism, not the opposite.

Page 13 line 314. So thermals are not turbulent?. Why do you separate rolls and steaks form thermals? Does it has to do with pollution transport or something similar?.

---

## Author Comment (AC1) · 21 Jul 2020

Thank you very much for your comments and for your thorough review. We addressed all of your comments and hopefully we will communicate better our methodology and the overall concept of our study.

Referee Comment: 1. In the manuscript a cosine fit is applied to wind lidar data acquired during VAD scans in order to estimate the mean wind speed and direction. Such a data analysis method requires the assumption of horizontal homogeneity of the wind vector, as also stated by the authors. However, I am wondering to which extent this homogeneity is expected over the urban landscape of Paris. I think that a discussion
on the terrain heterogeneity is missing from the article. In this direction, I think that it would be constructive to add an elevation map of the area over which the scanning wind lidar was acquiring measurements. This would provide an insight to which extent the observed spatial variations of the wind are related to temporal fluctuations of the wind and/or due to changes of the terrain elevation.

Authors' Response: 1. The lidar scans were covering the urban area of Paris as it can be seen in Figure 1b of the manuscript. In Paris there is a height limit for the buildings. Please see Figure 1 of the supplementary materials with the height limits of the buildings in Paris from the official Planning Department of the city of Paris, 2006. The Jussieu site is highlighted by the green text symbol. The buildings in the centre of Paris range between 25 and 37 m and rarely reach or exceed 50 m. So despite being an urban area, it is rather homogeneous. We will add the elevation map of the area. Please see Figure 2 of the supplementary materials. We also included the map with the altitude of the beam, see Figure 3 of the supplementary materials.

RC: In this context and regarding the data analysis: a. A statistical parameter is required to specify the representativeness of the fit used in the VAD scan. In the manuscript it is stated that the RMSE values of the estimated fit have been calculated but they are not stated in the document. b. Did the authors perform a quality check of the acquired data? Do they apply any SNR filtering to the acquired radial wind speed prior the application of the data analysis?

AR: a. The RMSE and similar statistical parameters for the fit quality evaluation incorporate in their values the difference between the radial wind speed and the fitted function which is the parameter we are interested in. Therefore, it is complicated to use such a parameter to characterize the quality of the fit. What we did instead, it was to select the bad cases based on the symmetry of the radial wind field. An example of a bad case is showcased in Figure 3a of the manuscript, where the radial wind field is not symmetric and as a result the radial wind speed shows a more chaotic behaviour when plotted against the azimuth angle (Figure 3, manuscript). On the contrary the good

cases were selected for symmetric radial wind fields as the one presented in Figure 2 of the manuscript. b. We forgot to include this information. We considered only the radial wind speed values for which the carrier-to-noise ratio (CNR) is higher than -27 dB. The values where CNR< -27 dB were filtered out since the radial wind speed had anomalously high values, 2 times or higher compared to the rest of the radial wind observations. In P4 Line 124 the following sentence will be added "The radial wind speed values for which the carrier-to-noise ratio is lower than -27dB (CNR<-27dB) were disregarded from the study since they were anomalously high, exceeding the values of the rest of the radial wind speed values by two times or higher.".

RC: 2. The subtraction of the mean wind speed from the radial wind speeds does not compensate the fact that the individual measurements along the scanning pattern are the result of the projection of the instantaneous wind vector to the line-of-sight of the wind lidar. Therefore, the term turbulent wind field could lead to a misinterpretation. The authors should clearly state that they measure the high frequency fluctuations of the radial wind speed.

AR: 2. It is true that using the word turbulent can be misleading since we do not observe the small scales. We will make clear in the manuscript that what we observe are medium-to-large fluctuations and coherent structures (mlf-cs). However, we will state that these are associated to a turbulent atmosphere.

RC: Specific Comments: P2 Line 28. The reference of Roth 2007 is a review of the atmospheric turbulence studies over urban landscapes and it doesn't directly discuss how the pollution concentration in urban environment is dependent on the weather and on the turbulence. A reformulation of this sentence is suggested for a clarification.

AR: The first paragraph will be removed along with the references. We will instead emphasize the effect of the turbulent structures in the pollutants' dispersion. In P1 L31-35 the following text will be added: "Several studies have been carried out to examine the effect of the coherent turbulent structures in the dispersion of pollutants by utilizing

boundary layer simulations. The results of these studies indicate that the turbulent structures can play a significant role in the pollutants' concentrations (Aouizerats et al., 2011; Soldati, 2005). Furthermore, Sandeepan et al., (2013) have demonstrated via simulations that the pollutants' concentrations can alternate from low to high during coherent turbulent structures events. It is therefore important to be able to identify turbulent structures in the atmosphere and observe them in an efficient and consistent way.".

RC: P3 Line74. Why is it relevant if the campaigns are short-term or long-term? And what is the time scale of these two types of campaigns that it is relevant for the topic of this study?

AR: We used an example of a short term study to show that so far lidars have been used to observe coherent structures for short periods since the analysis of a large dataset is very time-consuming. On the other hand, the example of a long-term study is mentioned to show that it provides us only point measurements instead of the whole wind filed over an area. Thus we wanted to highlight the benefits of our study. In order to clarify the limitations of the long-term study, we will add the following sentence after P3 line 77 "However, their study is limited to point measurements instead of the larger wind field we observe via lidars."

RC: P3 Line95. I am not sure that the two references of Kumer 2014 and Veselovskii et al. 2016 are the appropriate to be used here. In none of them there is an analytical explanation of the wind lidar instrument as it is stated. I suggest replacing them with more relevant references. An example could be the work of: Cariou, J. P., R. Parmentier, M. Valla, L. Sauvage, I. Antoniou, and M. Courtney. "An innovative and autonomous 1.5 $\mu$m Coherent lidar for PBL wind profiling." In Proceedings of 14th Coherent Laser Radar Conference. 2007.

AR: Thank you very much for this correction. After going through the references again we agree that Cariou et al, 2007 is the appropriate reference here, hence we replaced

the other two references by it.

RC: P3 Line 101. It is stated that “The duration of each scan was 3 minutes which is sufficiently short for the observation of structures”. Could the authors elaborate more on this statement?

AR: In P3 line 101 the word “fast” will replace the initial “short”. Furthermore, the sentence “The duration of each scan was 3 minutes which is sufficiently short for the observation of structures” will be rephrased to “The duration of each scan was 3 minutes which is sufficiently fast for the observation of coherent structures, as their lifespan is several minutes.”

RC: P4 Line 102. The authors state that the maximum range of the scans reached 5 km. However, in none of the figures that they have included in the manuscript the range ever reaches 5 km. The most common range in their data is between 2 – 2.5 km. Can the authors explain why is this happening?

AR: We have used the CNR < -27dB filtering. Please see also the reply 1b.

RC: P4 Line111. Table 2. A list of scanning patterns is included here that are not used in this study. In addition, the purpose of each scanning pattern is included without explaining the reasoning for their selection. I would suggest to either remove this table. If the authors wish to keep it then I suggest that they should elaborate more in the text about it.

AR: We acknowledge that the presentation of the entire scanning sequence can be confusing for the reader, therefore the Table 2 will be modified to include only the information for the PPI and the DBS scans. We have also removed the 90° elevation angle from the table as it was not part of the DBS scanning method. The latter scanning method is also important for this study as the DBS scans revealed the high wind shear cases during night. We will also change parts of the text. The sentence “Table 2 showcases the implemented scanning methods during the VEGILOT campaign that

are important for the current study” will replace the initial “Table 2 showcases the scanning sequence as it was implemented during the VEGILOT campaign.”. Additionally, a brief description of the DBS scan will be added to the text. In particular, the following sentence will be added in P4 line 103-106: “It was also important for this study to retrieve observations regarding the vertical wind shear. For this purpose, the Doppler beam swinging (DBS) scanning method was implemented. This method was consisted of four line of sight beams at azimuth angles of 0°, 90°, 180° and 270° with an elevation angle of 75° and it was applied twice. The duration of the four direction beams emission was approximately 15 seconds.”.

RC: P4 Line120. The “offset” refers to the vertical wind speed component? What does it mean that $\alpha$ is much smaller than b? Is this a common observation over the whole scanning plan? And what kind of $\alpha$ values does the application of the model result in?

AR: The offset is indeed associated to the vertical wind speed. The radial wind speed analysed in its’ wind components for a line of sight beam with azimuth angle theta and elevation angle phi will be: Vr=u*cos(theta)*sin(phi)+v*sin(theta)*sin(phi)+w*cos(phi), where u is the longitudinal, v is the transverse and w is the vertical wind component respectively. The offset is associated to the parameter w*cos(phi) (Thubois et al. 2018, Study of the configurations and scanning strategies of Doppler Lidars for providing wind and aerosol/cloud profiles). Since the elevation angle of the PPI scans was 1° and not 0° we included this parameter in the fit function. However, it is much smaller than the b component which is associated to the horizontal wind. So the horizontal wind is still dominant. This is a common observation over the whole scanning plan. The value of the offset is around 10 times smaller than the amplitude and ranges according to the value of the amplitude from 0.05 to 0.5 m/s.

RC: P6. Line156-157. It is not clear how the unaligned thermals are dependent on the increased solar radiation measurements.

AR: The increased solar radiation measurements result in surface heating which we

typically observe during fair weather cumuli conditions.

RC: P6. Line 163. How do the authors estimate the value of the wind shear? Also, shouldn't the units of the wind shear be seconds to the power of -1?

AR: The DBS observations provided us the zonal u and meridional v winds. The horizontal wind was computed from these components from the formula: V_hor=√(u^2+v^2). The wind shear was estimated from the vertical profile of V_hor by subtracting the local minima from the local maxima above it, near the surface. Thank you for this correction, the units have been corrected to s-1.

RC: P6. Line 166. The authors in P5. Line 133 state that the VAD method could not be applied in the data acquired during the night, especially at those occasions where the mean wind speed was less than 2 m/s. Does this mean that only cases with mean wind speed higher than 2 m/s were selected?

AR: The VAD method can be applied regardless of the mean wind speed values as long as the radial wind speed field is symmetric. It is true however that when the mean wind speed is lower than 2 m/s then the radial wind field has the risk to not be symmetric. For the training ensemble, regarding streaks, rolls and thermals only cases with moderate winds were selected (5-8 m/s) whereas for the category others, cases with weak winds below 2 m/s were included. We are currently preparing a separate study with the physics of the structures based on the whole classification from this methodology. The results show that we observe winds below 2 m/s mostly during the others category. More particularly the results are: 7 out of the 1145 streaks cases, 0 out of 420 rolls cases, 67 out of the 900 thermals cases and 670 out of the 2112 others cases.

RC: P6. Line 167. How well was the mean direction and speed estimated through the VAD method in the selected cases? I suggest adding a statistical parameter that describes the representativeness of the applied fit (equation 1) to the measured data (e.g. RMSE).

AR: As we previously stated statistical parameters as the RMSE are not appropriate for the evaluation of the fit since the interesting parameter for us is the difference between the observation and the fit. We instead selected the training ensemble based on how well symmetric or not was the radial wind field and by plotting random rings for each scan to confirm good fits as displayed in Figure 2 or a bad fit as in Figure 3 of the manuscript.

RC: P7. Line 170. It is difficult to understand the scale of the Modis colour images. Would it be possible to either add a scale or mark on the images the scanning area?

AR: In P7 line 170 the scanning area will be added to the MODIS images. See Figure 5 of the supplementary materials.

RC: P7. Line 176. I recommend describing very shortly the texture analysis especially in the context of remote sensing. It would be useful to add any references in the introduction regarding the previous applications of this type of analysis to remote sensing data.

AR: We will add the following brief description: "Texture analysis is an effective way to evaluate the distribution of the values within an image (Castellano et al. 2004). It is widely used in various scientific fields in order to classify images, covering meteorology (Alparone et al., 1990), medical studies (Holli et al., 2010) and forestry (Kayitakire et al. 2006).". If showing that the variety of the texture analysis in science is not appropriate, we can focus only on the remote sensing studies e.g. Alparone et al. (1990) and more.

RC: P8. Line 181-182. What is the logic behind the selection of this particular values for defining the contrast?

AR: We wanted to enhance the contrast of the structures. For this reason, we had to select the bins in such a way that the difference between positive and negative values will be more apparent. We have tried more configurations. The selection of only 2 bins (one positive, one negative), led to a less successful classification of the different

types of structures as the 2 bins give a 2 by 2 co-occurrence matrix. The size of the co-occurrence matrix is important in this case, since from the Equations 2, 3 and 4 of the manuscript, it is evident that the texture analysis parameters also depend on the distance between the bins i,j. The classification error in this case was around 18%. The selection of the 4 bins (one bin including all the negative values below −0.5 m/s, one bin between −0.5 m/s and 0, one bin between 0 and +0.5 m/s and one bin including all the positive values above +0.5 m/s) did not really improve the results with the error remaining around 18%. The selection of 8 bins reduced the error significantly. The selection of one bin including all the negative values below −0.5 m/s, six bins equally distributed between −0.5 m/s and +0.5 m/s and one bin including all the positive values above +0.5 m/s allows us to enhance the difference between the positive and negative values while keeping the distance between the bins i,j. We selected the values 0.5 as the limit in order to well separate the positive and negative values while having some information near 0. We also tested the limit with 1 and -1 m/s and the classification error was above 14%. We did not try to select more than 8 bins because we believe that it will not be useful to increase the number of bins near 0. It is in our future plans however to automatize this part as well. We would like to include an algorithm in our method in order to find the optimal selection of bins that minimizes the classification error.

RC: P10. Line 243. The authors state that 60 cases of “other” patterns are used during the supervised machine learning step. They argue that this is necessary because “it is expected to be the dominant category in the classification”. I am not sure that I understand what it is meant with this statement. I suggest having this part a bit more explained. Furthermore, how is the mean wind speed and direction estimated in these cases? As it is stated the VAD was not successfully applied to this data.

AR: When we were analysing the results, we observed that the chaotic type of patterns (see Figure 4 of the supplementary materials) was the most common type. The algorithm can be sensitive to an unbalanced training ensemble. It is preferable to select

a training ensemble based on the expected results (Kubat, 2017 p.194). Even for the bad cases, the VAD method was selected for the estimation of the mean wind speed and wind direction. In Figure_vi of the supplementary material, a fitted function for a non-symmetric field is displayed. We can still obtain the mean wind direction from this figure, but the patterns will have the chaotic look as in Figure 4 of the supplementary materials.

RC: P10. Line 225. How do the authors explain the change in the slope of the homogeneity curve that is observed for absolute azimuth angles larger than 45 degrees?

AR: The angle represents also the distance between two grid point. For 45° angles or above, the distance between two grid points are n rows away whereas below 45° they are n-1, n-2 etc. We have prepared an illustration of an ideal case, Figure 7 of the supplementary materials. We hope that it is clear. Keep in mind the Figure 6 of the manuscript refers to the third neighbour which is equivalent to 150 m distance between the grid points.

RC: P10. Line 240. Could the authors add a reference to the literature describing the "supervised machine learning methodology"?

AR: The following references will be added: "Bonamente, M. Statistics and analysis of scientific data, Springer, 2017, 318 p., DOI:10.1007/978-1-4939-6572-4; Gareth James, Daniela Witten, Trevor Hastie, Robert Tibshirani, An Introduction to Statistical Learning with Applications in R, Springer Texts in Statistics, 2013, 426 p, DOI: 10.1007/978-1-4614-7138-7 Kubat M. An Introduction to Machine Learning, Springer, 2017, https://doi.org/10.1007/978-3-319-63913-0"

RC: P11. Line 265. How do the authors physically explain this result? A low RMSE in the cosine fit, couldn't also mean that the mean wind speed and direction are not estimated correctly?

AR: This result show that according to the algorithm, the different shapes of the patterns in the radial turbulent wind field are more significant for the classification of the structures than the physical parameters. The wind values range from 1 m/s to 14 m/s. Therefore, a higher RMSE does not necessarily mean a worse fit, but it may be caused due to this scale difference.

RC: P12. Line 291. It is stated that “there were scarcely any rolls cases observed at night”. However, in the sentence 288 it is written that for the classification of this study the authors “consider only thermals and rolls during daytime”. To my understanding there is an inconsistency between the two statements. Can this be explained better?

AR: For the training ensemble only thermals and rolls during daytime were selected. However, the classification of all the data was made based on the five texture analysis parameters displayed in Figure 7 of the manuscript. As time is not included in the classifiers, the algorithm can classify the patterns at any time of the day but still few cases of rolls and thermals were classified during the night. This result is an indication that the classification is working as intended.

RC: P13. Line 312. In the conclusions the author state that time, wind speed and the cosine fit RMSE of the VAD method were not selected by the algorithm for the classification. However, in the results presented in figure 9 there is a time dependency in the detection of certain patterns (e.g. thermals and rolls). Could the authors comment why the inclusion of the time as a classification parameter would not improve further their results?

AR: The algorithm finds the best combination of parameters that minimize the classification error and time was not one of the five parameters as it can be seen in Figure 7 of the manuscript. By including the time as a parameter the classification error will not be reduced.

RC: P13. Line 318. Given the fact that one PPI scan lasts for 3 minutes and occurs every 18 means, can the authors explain how does the acquired data set contribute to the comprehension of the development of coherent structures?

AR: The lifespan of streaks can be several tens of minutes and for rolls it can be hours. Even with this time gap between the observations we believe that it is still interesting to study the transitions between the different types of structures. For example, we have vertical lidar observations between the PPI scans and we can study the development of the atmospheric boundary layer height with regards to the type of the structures.

RC: Technical Corrections: General: There is a small inconsistency on the way that figures are referred to. Sometimes they are used parenthesis after the number of the Figure to denote a subfigure and sometimes are not.

AR: All the subfigures will be corrected in order to keep consistent naming. The number of the subfigure will be followed by the corresponding letter e.g. “Figure 1a”.

RC: P2 Line 34. I think that the statement “Futhermore, . . . ..” is a self-evident. I would suggest removing it. Also, I would suggest to add the reference of (Hussain, 1983) at the end of the previous sentence.

AR: In P2 line 34 the sentence ”Furthermore . . ... time-averaged statistics calculations.” will be removed. The reference of (Hussain, 1983) will now follow the sentence “The principal aspect that determines a coherent structure is the maintenance of the phase-averaged vorticity of the turbulent fluid mass over the spatial extend of the flow structure.”.

RC: P2 Line 37. “and the lower”

AR: In P2 line 3 “and the lower” will replace “and lower”.

RC: P3 Line 72. I suggest reformulating this sentence. It is not clear to the reader how is the two-dimensional autocorrelation function was used. Also, I suggest changing the sentence “the observation of the scans by eye” to “visual observation of the scans”.

AR: In P3 line 72 the sentence “They combined quantitative characteristics of the coherence such as the integral scales and the anisotropy coefficients, obtained by a two-dimensional autocorrelation algorithm, with the visual observation of the scans.”

will replace the initial “They combined a two-dimensional autocorrelation function with the observation of the scans by eye”.

RC: P3 Line76. Change “month” to “months”

AR: In P3 line 76 “months” will replace “month”.

RC: P3 Line83. The “Section 0” should change to “Section 3”

AR: In P3 line 83 “Section 3” will replace “Section 0”.

RC: P3 Line86. The word “two” is used as a noun modifier and therefore the word month should be in singular form.

AR: In P3 line 86 the phrase “A two-month measurement campaign” will replace the initial “A two-months campaign”.

RC: P3 Line96. I suggest to change the text “The lidar is sensitive only to the” to “A wind lidar is measuring the”

AR: In P3 line 96 the phrase “A wind lidar is measuring the” will replace the initial “The lidar is sensitive only to the”.

RC: P3 Line100. Change the “for a” to “with a”.

AR: In P3 line 100 “with a” will replace the initial “for a”.

RC: P4 Line103. I think that it is more grammatically correct to either use the past tense or the passive form of the “rise” verb.

AR: In P4 line 103 “the beam rise” will be rephrased to “the beam was risen”.

RC: P5 Line132. The authors state that due to the surface heterogeneity the VAD method can be applied in some cases. A surface heterogeneity will introduce an error in the VAD method regardless the wind speed.

AR: We have probably phrased this in a wrong way. As it can be seen in Figure 2 in the

supplementary material, there are some hills in the limits of the scanning range but with low elevation. We have not study what could be the orographic effect in the wind speed as this would require model simulations. However, there have been relevant studies for the area. Troude et al. (2002) in their study “Relative influence of urban and orographic effects for low wind conditions in the Paris area” and Lemonsu and Masson (2002) in their study “Simulation of a summer urban breeze over Paris” examined the orographic effect of the nearby hills as well as the urban heat island effect in the synoptic wind speed over Paris. The results show that during low wind conditions these effects can be the main driving forces for the local wind speed. We did not include these references in our study because they refer to spring-time and summer-time conditions respectively and only for case studies. We are not aware of a longer term or an autumn study.

RC: P5 Line132. I suggest that the “Jussie site” is changed to “The Jussie site”.

AR: In P5 line 132 “The Jussieu site” will replace “Jussie site”.

RC: P5 Line138. Figure 3 caption. Change the “a case” to “A case”, also a add a tab space between (a) and “Radial”.

AR: In P5 line 138 “A case” will replace “a case” and a tab space will be inserted between (a) and “Radial” in the caption of Figure 3 of the manuscript.

RC: P6. Line149. “Sec” should be replaced by “Section”.

AR: In P6 line 149 “Section” will replace “Sec”.

RC: P6. Line 164. I suggest to re-write the sentence “For many cases, the wind shear was accompanied by turbulent streaks pattern” and specify for which particular wind shear values were streaks detected. RC: P6. Line165. The part of the sentence “so or the training ensemble” should be rewritten.

AR: In P6 line 164 the sentence “Therefore for the training ensemble, only night cases when streaks patterns (Figure 4c, manuscript) were accompanied by wind shear higher than 2 m/s near the surface, were selected.” will replace the initial “For many cases,

the wind shear was accompanied by turbulent streaks patterns (Figure 4c, manuscript) so or the training ensemble, only night cases of streaks were selected to ensure that wind shear was the primary factor for the generation of turbulence.”.

RC: P10. Line 230. “fore” should be changed to “for”.

AR: In P10 line 230 the word “for” will replace the initial “fore”.

RC: P11. Line251 – 253. The authors state the higher number of dimensions relative to the number of patterns lead to the “curse of the dimensionality problem”. I suggest to re-write it by using appropriate scientific statements.

AR: The "curse of the dimensionality problem" is a scientific term, which is common in statistics/data science domain. We think that it is relevant here.

RC: P11. Line 259. Correct the “Section 0”.

AR: In P11 line 259 “Section 3” will replace “Section 0”.

RC: P12. Table 5 caption: I suggest changing the “eye-made” to “visual”.

AR: In P12 the word “visual” will replace the initial “eye-made” in the caption of Table 5.

RC: P12. Line 290. The coherent structures don’t have a preference. They are formed under favourable atmospheric conditions. I suggest commenting the result of Figure 9 on that basis.

AR: This was undoubtedly a bad way to phrase it. We will make the following change in the text: “It is evident that despite time was not one of the selected classifiers, the number of occurrences of the structures show a distribution that can be associated to the atmospheric conditions. More particularly, rolls and thermals were mainly classified during day. This result is noteworthy as these structures are linked to a well-developed atmospheric boundary layer during the day. On the contrary there were scarcely any rolls cases observed at night, and a few unaligned thermals were classified at night.” will replace the initial “It is evident that despite time was a much less significant classifier compared to the curves parameters, the structures show a time preference. There were scarcely any rolls cases observed at night, though a few unaligned thermals were classified at night."

RC: P15. Line 386. "Lemone" should be changed to "LeMone". Also, necessary information of the article (e.g. journal) is missing.

AR: In P15 line 386 the name will be corrected and the missing information will be added. The complete reference now is: "LeMone, M., 1972. The structure and dynamics of the horizontal roll vortices in the planetary boundary layer. J. Atmos. Sci. 30, 1077–1091. https://doi.org/10.1175/1520-0469(1973)030<1077:tsadoh>2.0.co;2"
* * *
[Figure]

E - HAUTEURS
Zones Urbaines
Hauteur plafond en mètre
18 25 31 37
50 180
Territoires couverts par les fuseaux de protection du site de Paris
Zone Naturelle et forestière
Echelle : 1/25 000ème
0 500 1 000 1 500 M
PLU modifié les 6 et 7 février 2012
Plan de Sauvegarde et de Mise en Valeur du 7ème arrondissement
Plan de Sauvegarde et de Mise en Valeur du Marais
1er
2ème
3ème
4ème
5ème
6ème
7ème
8ème
9ème
10ème
11ème
12ème
13ème
14ème
15ème
16ème
17ème
18ème
19ème
20ème

Fig. 1.

[Figure]

**Fig. 2.**

[Figure]

**Fig. 3.**

[Figure]

-1
-2
-3
-4
-5
-5 -4 -3 -2 -1 0 1 2 3 4 5
Distance (km)
1
0.5
0
-0.5
-1
Turbulent radial wind speed (m/s)

Fig. 4.

[Figure]

**Fig. 5.**

[Figure]

Cosine fit at 11:38:50
observations
Radial wind speed (m/s)
1
0.5
0
-0.5
-1
-1.5

Azimuth angle

**Fig. 6.**

[Figure]

Texture parameters: co-occurrence matrices
Ideal wind field
24x24 grid points
Cell pairs aligned with the mean wind
Co-occurrence matrix for
first neighbour points at 90°:
Wind bin for cell #2
Wind bin
for cell #1
Number of
cell pairs
with [5,2]
values
1 2 3 4 5 6 7 8
1 69 0 0 0 0 0 0 0
2 0 69 0 0 0 0 0 0
3 0 0 69 0 0 0 0 0
4 0 0 0 69 0 0 0 0
5 0 0 0 0 69 0 0 0
6 0 0 0 0 0 69 0 0
7 0 0 0 0 0 0 69 0
8 0 0 0 0 0 0 0 69
Examples of cell pairs perpendicular to mean wind
spaced by 1
period
spaced by ½
period
Co-occurrence matrices are
computed for:
• Various distances,
i.e. neighbour orders
n from 1 to 30
(50 m to 1.5 km)
• All possible cell pair
orientations,
i.e. azimuth φ
from 0 to 180°
Depending on the angle the
two neighbour points can be
closer or further. For angles
larger than 45° the distance is
smaller and thus it is more
likely to observe neighbour
points of the same bin. Hence
it is possible to observe a
slope in a texture parameter-
azimuth angle figure.
φ
0°
45°
67.5°
90°

**Fig. 7.**

---

## Author Comment (AC2) · 21 Jul 2020

Thank you very much for your comments and for your analytical review. We hope that by addressing your questions we will clarify the aim of this study, as well as our methodology.

Referee comment: As I understand form the first paragraph in the introduction, comprehension of the flow physics it is important for monitoring atmospheric pollution. However, the physics identified here, in the form of coherent structures, are not related with bad pollution conditions, since the latter might fall in the "Other" category, and their physics are not clear from the study. The motivation of the study should be stressed

form the beginning. Only when we arrive to the conclusions we can read some of the potential application of this approach. (line 316 of the text). There are many previous studies on coherent structures, as well as lidar technology, than could be included. The description of the coherent structures that this study aims to identify is a bit vague and needs improvement. In general, the term turbulence and turbulence fields are used frequently in the text, but the range gate resolution of the lidar scanner is 50m at a height of 75m above ground level. Turbulence and its most energetic eddies might fall within this length scale. Almost all turbulence fluctuations are filtered out by the lidar due to spatial averaging. What we can clearly see from lidar observations is medium-to-large scale fluctuations and coherent structures rather than turbulence. The methodology behind data process could be better explained. I miss a paragraph describing how data quality was assured.

Authors' response: We recognize that the first paragraph can be misleading and therefore it will be removed. We will focus only on the effect of turbulent structures in the pollutants' dispersion. In P1 L31-35 the following text will be added: "Several studies have been carried out to examine the effect of the coherent turbulent structures in the dispersion of pollutants by utilizing boundary layer simulations. The results of these studies indicate that the turbulent structures can play a significant role in the pollutants' concentrations (Aouizerats et al., 2011; Soldati, 2005). Furthermore, Sandeepan et al., (2013) have demonstrated via simulations that the pollutants' concentrations can alternate from low to high during coherent turbulent structures events. It is therefore important to be able to identify turbulent structures in the atmosphere and observe them in an efficient and consistent way.". We want to show that it is possible to identify and classify these structures based solely on the patterns from the fluctuations of the radial wind speed data by combining texture analysis and supervised machine learning. For this reason, we will add the following text in the last paragraph: "This study aims to identify the coherent structures on single Doppler lidar horizontal scans and develop an automatic classification process based on the combination of texture analysis and a machine learning technique in order to handle large datasets. There is a lack of

long-term studies of coherent structures based on lidar observations and the aforementioned automatic classification process can stimulate the interest in this research field. More particularly, it could facilitate the statistical analysis of the physical parameters of the structures, e. g. the structure size as a function of the planetary boundary layer (PBL) height. Furthermore, it will enable us to study the transitions between structures and how these are associated to the atmospheric conditions. Finally, the impact of the coherent structures on pollutants' dispersion could be examined for long-term studies under stable and unstable conditions." We will replace the term turbulence to the suggested medium-to-large fluctuations and coherent structures (mlf-cs) as we do not observe small scale turbulence and the reader can be confused. Nonetheless, we will state that these stuctures are associated to a turbulent atmosphere. Unfortunately, during the VEGILOT campaign there was no other wind data measurements for comparison. The closest weather station with available data for the same period, as our study, is located in Montsouris, 20 km away from the centre of Paris.

RC: Did you use a CNR/SNR threshold for filtering?

AR: We used the CNR filtering. The radial wind speed values below -27 dB were anomalously high and therefore excluded from the computations. In P4 Line 124 the following sentence will be added "The radial wind speed values for which the carrier-to-noise ratio is lower than -27dB (CNR<-27dB) are disregarded from the study since they were anomalously high, exceeding the values of the rest of the radial wind speed values by two times or higher.".

RC: What was the data recovery rate during the two months of measurements?

AR: The lidar was taking measurements continuously for the two-month period for the scanning sequence presented in Table 2 of the manuscript. There was no pausing of the measurements during this period.

RC: It is not clear from the text how the radial wind speed fluctuations are calculated. What does "stronger radial wind speed" mean? A larger absolute value? It seems to

me that the sign might come from the combination of u and cos ($\theta$). One suggestion is to put this definition as equations.

AR: The radial wind field has values with a positive sign when the wind moves away from the lidar and negative sign when it moves toward the lidar. In order to be consistent throughout the field when we study the fluctuations, we have to make a sign convention that guarantees for example that the radial wind speed with a value of 6 m/s and the radial wind speed with a value of -6 m/s will be part of the same pattern. Therefore, we compared the absolute values and we derive the sign convention of Figure 2b of the manuscript. We will add the following equation in the text as well: u_r'=|u_r ($\theta$)|-|f($\theta$)|, where f is the fitted function for the corresponding azimuth angle $\theta$.

RC: Since the wind direction is obtained from equation (1) it would be possible to work with the streamwise component u instead of the radial wind speed ur. It is not clear from the results, what is the relative importance of non-distinguishable structures, bad fittings, and bad data (low CNR/SNR signals) in the "Others" category. The authors give some information about what it seems to be the reason of one group of cases to belong to this category (bad fitting of cosine function), but no threshold on this fitting error is given.

AR: We did not perform an extensive analysis of the data for the "Others" category. This category was selected exclusively to separate the not interesting patterns. The bad cases were not selected based on the quality of the fit but rather in the symmetry of the whole radial wind field. For the category "others" in the training ensemble, we selected 53 bad cases with an asymmetrical radial wind field and 7 cases of a symmetrical field where the structures were non-distinguishable. In Figure 3a of the manuscript, it is apparent that the wind field is not symmetrical and that the VAD method would not be applied efficiently. Nevertheless, it can still be applied as you can see in Figure 4 of the supplementary materials (it is the same as Figure 3b of the manuscript along with a cosine fitted function). Since we are interested in the fluctuations (difference between the observation and the fitted curve) it is very difficult to characterize the quality of the

fit by using a parameter such as the RMSE that contains these fluctuations. We are currently working on a study for the physical parameters of the structures based on the classification of this study. We found that for 670 out of the 2112 others cases the mean wind speed is lower than 2 m/s. On the other hand, for the streaks it is only 7 out of the 1145 cases, for rolls 0 out of 420 cases and for thermals 67 out of the 900 cases. This is very interesting as the mean wind speed was not one of the classifiers.

RC: I miss more elaboration in the description of the texture parameters used for classification, namely, why they might be relevant and if they were relevant in the end. The feature selection process is also not very clear. Cross validation is well known and well explained, but the text explaining the outcome as well as the figure used in that regard are confusing. A brief description of the machine learning technique used could be useful for clarity.

AR: The description of the texture parameters will be added in the manuscript, in particular: "Correlation indicates the existence of linear structures in the image, with high values associated to a large amount of linear structure in the image. Contrast reveals the local variations in an image, where a large amount of variation leads to high values. Homogeneity is self-explanatory and the high values represent a homogeneous image. Finally, energy measures the uniformity of an image with the highest values corresponding to constant or periodic forms.". These texture parameters are frequently used in patterns based image classification. As we mention in our manuscript we selected these four parameters inspired by the study of Srivastava et al. (2018). They used the same parameters to distinguish stripes among others patterns. By plotting the texture parameters against the azimuth angle $\varphi$ (angle of the comparing neighbour points), we observed a prominent peak for the elongated patters when $\varphi$ is equal to the mean wind direction. The relevant parameters are showcased in Figure 7 of the manuscript. Regarding the feature selection process by the algorithm the following text will be added: "The greedy algorithm of stepwise forward selection was used in the article, which is the standard and frequently used method of reduction of the feature space. As indicated in (Sokolov et al., 2020), the feature selection is an iterative procedure when features are divided into two groups - accepted in the classification model and remaining. Features from the second groupset of “remains” are successively added to the model and corresponding classification errors are estimated. The minimum is chosen from the set of error estimations and compared with the error of the previous model, which is based on previously accepted features. If the error reduces significantly, then the corresponding feature is included into the updated model. If the error is not diminishing significantly for any feature from the remaining group then the process stops.” For the specific machine learning technique we utilized, the following will be added in the manuscript: “Quadratic discriminant analysis or normal Bayesian classification (see Hastie et al. 2008) is the parametric approach implying that probability density functions (PDF) belong to the family of normal distributions. It is a classical algorithm of the supervised machine learning, based on the principle of maximum likelihood. The general idea is to estimate the PDF for each class, and then select the most probable class (see Kubat 2017).”

RC:Conclusions section. In my opinion, this section should be read in a positive rather than negative way. Example: it should focus on the relevant parameters discovered (which need a bit more explanation in the corresponding section) rather than the ones excluded by the study. The sections describing the methodology used-which need some improvement-are already clear, and no repetition is needed. Same with the results highlighted.

AR: It is very important for us to stress the positives of this study and therefore we will modify the conclusions to comment on the relevant parameters. The following text will be added in the main part of the manuscript and restated in a modified version in the conclusions: ”The algorithm allowed classifying correctly about 91% of the dataset using five texture analysis parameters as predictors. Analytically these parameters are the amplitude of the 2nd-neighbour homogeneity curve, the integral of the 18th-neihgbour contrast curve, the amplitude of the 4th-neighbour contrast curve, the

integral of the 8th-neighbour correlation curve and the symmetry of the 2nd-neigbour homogeneity curve. These results show that the prominent peaks are a distinctive characteristic for the elongated patterns as the amplitude of the homogeneity and contrast curves are two of the significant parameters. Furthermore, the integral or more precisely the sum of the points of the curves for the contrast and for the correlation curves are significant parameters as well. This is important especially for the distinction between the categories thermals and "others" as their amplitude may not differ substantially since the patterns are not towards a specific direction, yet a chaotic area will have higher values of contrast and lower values of correlation compared to an enclosed homogeneous area. Finally, the symmetry of the homogeneity curve as a classifier reveal the urgency to align the radial turbulent wind fields to the mean wind direction and thus align the structures such as streaks and rolls with the mean wind direction in order to be distinguishable from the random positions of the enclosed structures of the thermals or the chaotic structures of the "others". It is also crucial to note that the parameters cover various distances, from the 2nd-neighbour, which in grid points is 100 m to the 18th-neighbour which is 900 m. This is necessary for our classification since streaks and rolls are both elongated patterns but their transverse horizontal sizes differ."

RC: 2 Specific comments Page 1 Line 12. Change "manually" to "visually".

AR: In P1 line 12 the word "visually" will replace the initial "manually".

RC: Page 1 Line 15. Change "and installed" to "installed".

AR: In P1 line 15 the word "and" will be removed with the phrase now reading "by a scanning Doppler lidar (LEOSPHERE WLS100) installed".

Page 1 Line 16. It would be better to reword this sentence, maybe "The turbulent component of radial wind speed is estimated using. . .over 4577 scans.

In P1 line 16 the sentence "The lidar recorded 4577 quasi-horizontal scans for which the turbulent component of the radial wind speed was determined using the velocity azimuth display method." will be rephrased to "The turbulent component of radial wind speed is estimated using the velocity azimuth display method over 4577 quasi-horizontal scans.".

RC: Page 1 Line 18-21. I am not sure what the sentence describing the training set adds to the abstract if not combined with the next one. It is better to state directly the unsupervised algorithm used instead of using parenthesis. It might be better to rephrase this in a more concise way.

AR: In P1 lines 17-21 the text "The differences between the three types of structures were highlighted by enhancing the contrast of the images and computing four texture parameters (correlation, contrast, homogeneity and energy) that were provided to the supervised machine learning algorithm (quadratic discriminate analysis)." will be rephrased to "The differences between the three types of structures were highlighted by enhancing the contrast of the images and computing four texture parameters (correlation, contrast, homogeneity and energy) that were provided to the supervised machine learning algorithm, namely the quadratic discriminate analysis. The algorithm was able to classify successfully about 91% of the cases based solely on the texture analysis parameters. In particular, the algorithm performed best for the streaks structures with a classification error equivalent to 3.3%.".

RC: Page 1 Line 23. What are the remaining 20

AR: The remaining 20% are the unaligned thermals. We wanted to highlight the results only for the rolls and streaks as we find in literature that the majority of the studies focus on these two types of coherent structures.

RC: Page 2 Line 26. Change "step for" to "step towards".

AR: We will remove the first paragraph of the introduction and instead address only the impact of turbulent structures on pollutants' dispersion.

RC: Page 2 Line 32. A coherent structure is defined according to its phase-averaged

rather than its instantaneous vorticity. I also suggest moving the Hussain 1983 reference to this sentence. A coherent structure needs to maintain its phase-averaged vorticity rather than its time-averaged vorticity or form.

AR: Thank you very much for this comment. In P2 line 32 the adverb “instantaneously” was referring to the phase-averaged vorticity. In order to avoid confusion, the text in P2 lines 32-34: “The principal aspect that determines a coherent structure is the instantaneously space and phase correlated vorticity of the turbulent fluid mass over the spatial extend of the flow structure. Furthermore, a coherent structure must maintain its form for a time period sufficient for time-averaged statistics calculations (Hussain, 1983).” will be rephrased to “The principal aspect that determines a coherent structure is the maintenance of the phase-averaged vorticity of the turbulent fluid mass over the spatial extend of the flow structure (Hussain, 1983).”.

RC: Page 2 Line 35. Please specify that this is the case of atmospheric flow. Other structures are observed at laboratory scale (also in the atmosphere but not so relevant for momentum or scalar fluxes), like hairpins, or hairpin trains. Include a reference to Hutchins and Marusic (2006) and Adrian (2007).

AR: It is definitely a good idea to specify that this study is related to the atmospheric flow. We will briefly mention laboratory experiments including references such as Hutchins and Marusic (2006) and Adrian (2007).

RC: Page 2 lines 37-44. Consider reordering the sentences here for a more fluent reading. Maybe starting the paragraph from sentence in line 41?

AR: It is true that by moving the sentence of line 41 to the beginning, the text becomes more fluent so we will apply this suggestion.

RC: Page 2 line 45. How is it that you identify rolls in the mixed layer, with sizes from few to dozen kilometers, with scans at surface layer height (75m) with spatial coverage of less than 2 kilometers?. Is this description coherent with what you are identifying?

AR: In the introduction we give general information about the rolls and we think it is important to address their scale. We will state in our data that the structures we observe are near the surface so at the base of the rolls.

RC: Page 2 lines 57-65. Since you are using lidar instead of radars it would be better to shorten the scanning pattern description using some of the given references, since they have to do only with the history on scanning patterns. There are more recent references of this regarding lidars. Cariou (2007) and Vasiljevic (2016).

AR: Thank you very much for the suggested references. We will include them in our study along with other studies for turbulent structures, such as Newsom et al (2008), Lin et al (2008) and more. The Cariou (2007) reference will also replace the initial "Kumer et al. (2014) and Veselovskii et al. (2016)" references as we realized that it is more relevant for the sentence: "The Doppler shift frequency between the emitted laser beam and the light backscattered by the aerosols is measured by heterodyne detection associated with Fast Fourier Transform as explained analytically by Cariou et al (2007)".

RC: Page 3 line 72. Replace "by eye" by "visual inspection" or similar.

AR: In P3 line 72 the phrase "visual observation" will replace the initial "by eye".

RC: Page 3 line 73. More than time-consuming, it might be non-systematic.

AR: This is a very powerful way to phrase it in order to show why it is important to develop an automatic method for the classification. We will include this term in our manuscript.

RC: Page 3 line 74. You meant "A less time-consuming" approach? What height was the met mast?

AR: The term less expensive refers to the sonic anemometers themselves without including the met mast. The data for the Barthlott et al (2007) study was taken at a 30 m tower.

RC: Page 3 line 78. “This study aims to identify turbulent coherent structures from single Doppler lidar horizontal scans”. Also, please introduce here what is texture analysis (roughly maybe) and what what machine learning technique you are using.

AR: Regarding the texture analysis, we will add the following text: ”Texture analysis is an effective way to evaluate the distribution of the values within an image (Castellano et al. 2004). It is widely used for varying scientific fields in order to classify images, covering meteorology (Alparone et al., 1990), medical studies (Holli et al., 2010) and forestry (Kayitakire et al. 2006).” For the machine learning technique, we will add the following text: “Quadratic discriminant analysis or normal Bayesian classification (see Hastie et al. 2008) is the parametric approach implying that probability density functions (PDF) belong to the family of normal distributions. It is a classical algorithm of the supervised machine learning, based on the principle of maximum likelihood. The general idea is to estimate the PDF for each class, and then select the most probable class (see Kubat 2017).”

RC: Page 3 line 83. Section 0 must read section 3, here and in the rest of the text.

AR: “Section 3” will replace “Section 0” throughout the text.

RC: Page 3 line 86-91. It should read “measurement campaign”. Move “in Paris” to the end of the sentence, modified to “in the urban area of Paris”. Remove the url of leosphere to the reference section maybe. More than only sensitive to the radial component, the lidar does measure and it is intended to measure only the radial component. Lidars technology and its operation principle is well known, use references (Cariou, maybe write ts paragraph in amore concise way, being specific in the corresponding table than in the text here, which is a bit confusing.

AR: In P3 line 86 the phrase “A two-month measurement campaign” will replace the initial “A two-months campaign”. The phrase “in the urban area of Paris” will also be inserted in P3 line 88. The Cariou (2007) reference will also replace the initial “Kumer et al. (2014) and Veselovskii et al. (2016)”.

RC: Page 3 line 101. I would say fast instead of short. Also, what type of structures and why this time window?

AR: We refer to coherent structures, in P3 line 101 the sentence "The duration of each scan was 3 minutes which is sufficiently fast for the observation of coherent structures, as their lifespan is several minutes." will replace the initial "The duration of each scan was 3 minutes which is sufficiently short for the observation of structures.". The time-window of 3 min is the result of the selection of the 2° azimuth angle resolution. We wanted to combine a high spatial resolution with a time-window that would allow us to observe coherent structures.

RC: Page 4 line 121. What is the reason behind a is small for your case?.

AR: The parameter a is associated to the vertical wind speed, more particularly with the parameter w*cos(phi) of the radial wind speed analysed in its' wind components for a line of sight beam with azimuth angle theta and elevation angle phi: Vr=u*cos(theta)*sin(phi)+v*sin(theta)*sin(phi)+w*cos(phi), where u is the longitudinal, v is the transverse and w is the vertical wind component respectively. The offset is associated to the parameter w*cos(phi) (Thubois et al. 2018, Study of the configurations and scanning strategies of Doppler Lidars for providing wind and aerosol/cloud profiles). In our case the elevation angle is only 1° thus a is around 10 times smaller than the parameter b throughout our data.

RC: Page 5 Figure 2. Why the reach of the lidar is 2 km and no 5 km?.

AR: Due to the CNR<-27dB filtering.

RC: Page 5 line 134. Is it possible to specify the fraction of cases with low winds, and its relative importance to the number of bad fittings?. I miss an analysis of stability conditions, since it seems that stable conditions affect the most.

AR: As we mentioned previously, we examine the symmetry of the radial wind field rather than the individual bad fit for each ring. We are currently working on a separate

study for the physical properties of the structures and the atmospheric conditions under their occurrences. We are interested in this result as well, however we have not finished the study yet. Nonetheless, the preliminary results show that low winds (<2 m/s) are the main cause for non-symmetric radial wind fields.

RC: Page 6 line 143. Actually, for rolls, it is the opposite. Ascending motions bring low momentum to higher levels, reducing the speed, and vice versa.

AR: This was a mistake. It will be corrected.

RC: Page 6 line 146. Since rolls and streaks both present areas of alternating low/high momentum with elongated shape, their main difference is their extent. What is the criteria to differentiate between them? The clouds formation shape from MODIS was used, as I understand, only for a fraction of the cases included in the training dataset.

AR: For the training ensemble we combined the patterns of the fluctuations of the radial wind speed field with physical characteristics that indicate the existence of a structure. For streaks we selected cases with wind shear higher than 2 s-1 near the surface and for rolls, cases when clouds streets were formed over Paris as observed from MODIS satellite images. Due to the scarcity of satellite data, in order to select 30 cases of rolls we also included the consecutive cases of the cloud streets ones, as long as the patterns persisted.

RC: Page 6 line 161-165. Wind shear is defined as du/dz with 1/s units, could you clarify what definition you are using here? Additionally, streaks are present in turbulent flow as well, beyond stable conditions, why do you focus in cases with low turbulence energy (stable conditions)? It seems that high shear due to jets is only one among several mechanisms.

AR: The units will be corrected to 1/s. For the computation of the horizontal wind, we used the DBS observations. In particular the horizontal wind was computed by the formula: V_hor=√(uˆ2+vˆ2), where u is the zonal and v is the meridional winds.

The wind shear was estimated from the vertical profile of V_hor. We only used night streaks because the wind shear is a clear indication for the existence of streaks. As the algorithm only uses the five texture analysis parameters for the classification (Figure 7 of the manuscript), it shouldn't affect the results.

RC: Page 6 line 166. How many cases did you use for the "Other" category? From table 5 seems that they are around 60, the double. What is the reason for such big number?. Can this influence the final classification output? This is explained in section 4, but it should be clear from here.

AR: We used 60 cases which is double compared to the rest. The reason is that some of the machine learning algorithms are sensitive to the balance of classes in the training data (see Kubat 2017, p 194). If one category is dominant but for the training ensemble all categories are represented by the same number of cases, then the algorithm can overestimate or underestimate a category. Nevertheless, we also tried a training ensemble with all the categories represented by 30 cases and the results were similar.

RC: Page 7 Figure 4. What is the scale of map in (d) and (e)?

AR: We will add the scanning area in the images. Please see Figure 5 of the supplementary material.

RC: Page 7 line 176. Could you introduce what texture analysis is first? Additionally, since "Others" had a poor fitting and then uncertain wind direction, how did you align them with 0 degrees?

AR: As we mentioned above we will include a brief description of the texture analysis in the introduction. The VAD method was used even for the bad cases, as the radial wind field in this case fell in the category of not interesting. Please see Figure 6 of the supplementary materials where it is still possible to fit a cosine function even for a bad fit and how the radial turbulent wind field looks in that case in Figure 4 of the supplementary materials.

RC: Page 7 line 180. Eight bins were chosen for increased contrast. Why eight?, could you develop more on this?. What is the effect of the number of bins in the output?

AR: The scope was to enhance the contrast of the structures for a better visualization of the alternating positive and negative areas in the radial turbulent wind field. The selection of only 2 bins (one positive, one negative) was not very successful and gave a classification error of approximately 18%. In this case the co-occurrence matrix was 2 by 2. It apparent from Equations 2,3 and 4 that the distance between the bins i,j hence the algorithm could not classify the structures quite successfully based on the texture analysis parameters. Similarly, with the selection of the 4 bins (one bin including all the negative values below $-0.5$ m/s, one bin between $-0.5$ m/s and 0, one bin between 0 and +0.5 m/s and one bin including all the positive values above +0.5 m/s), we did not really manage to improve the results with the error remaining around 18%. Only when we selected 8 bins we succeeded in reducing the error significantly. The selection of one bin including all the negative values below $-0.5$ m/s, six bins equally distributed between $-0.5$ m/s and +0.5 m/s and one bin including all the positive values above +0.5 m/s allows us to enhance the difference between the positive and negative values while keeping the distance between the bins i,j. We selected the values 0.5 as the limit in order to well separate the positive and negative values while having some information near 0. We also tested the limit with 1 and -1 m/s and the classification error was above 14%. We did not increase the number of bins to more than 8 bins because we think that the increase of the number of bins near 0 will not improve the classification error. However, we are interested in the future to develop an algorithm that finds the optimal selection of the bins' limits in order to minimize the classification error.

RC: Page 8 line 185. The procedure for the construction of the CM matrix is a bit confusing. Could you write it in a more concise way?

AR: In P8 line 188 the following sentence "The rows and columns of the CM represent the different wind levels from 1 to 8, whereas the cells contain the frequency of the combination of two neighbour pixels in the image" will replace the initial "The rows and

columns of the CM represent the wind levels from 1 to 8, whereas the cells contain the number of occurrences of neighbour pairs with values corresponding to the row and column index.”.

RC: Page 9 line 212. Is it possible to elaborate more on the 4 parameters described?. It is not clear only from the equations what their characteristics are.

AR: The following text will be added in the manuscript: “Correlation indicates the existence of linear structures in the image, with high values associated to a large amount of linear structure in the image. Contrast reveals the local variations in an image, where a large amount of variation leads to high values. Homogeneity is self-explanatory and the high values represent a homogeneous image. Finally, energy measures the uniformity of an image with the highest values corresponding to constant or periodic forms.”.

RC: Page 10 Figure 6. The notation of the azimuth angle is different form the text. Why does homogeneity grow after 45 degrees for all categories? The definition of homogeneity says that CMs with large values in the diagonal might result in larger values of this parameter. The diagonal from table 3 to 4 decrease because of azimuth angle. Should homogeneity decrease monotonically from 0 to +/- 90 degrees?. Can you elaborate more on this?. How many cases are represented for each category in the figure? Only one scan? An average from many cases?

AR: The notation of the azimuth angle in the y axis of the Figure 6 will be corrected. The angle also represents the distance between two grid point. For 45° angles or above, the distance between two grid points are n rows whereas below 45° they are n-1, n-2 etc. We have included a pdf file in the supplementary material where we showcase an ideal case. We hope it is clear. Regarding your question whether homogeneity should decrease monotonically from 0 to +/- 90 degrees, the response is that it depends on the case and on the order of the neighbour. When we have elongated patterns then yes we can see the prominent peak at 0° as we see in Figure 6 of the manuscript for the streaks and to some degree for the rolls but not so much for the thermals and the

others. Please see also Figure 7 of the supplementary materials where we showcase an ideal case and how the values of the co-occurrence matrix change according to the periodicity of the patterns.

RC: Page 10 line 231. Notation is a bit weird here.

AR: The maximum and minimum refer to the azimuth angle $\varphi$. Maybe it would be more clear if written as follows: Hom.Amp(n)=max_$\varphi$(Hom($\varphi$,n))-min_$\varphi$(Hom($\varphi$,n))

RC: Page 10 line 241. The description of the training set might be better place in section 3. Why is it expected that "other" category should double the rest? Please elaborate more on this.

AR: The preliminary analysis of patterns showed that "others" class is approximately twice abounded than each other class. We decided to double the number of examples "others" class in the training dataset, as some of machine learning algorithms are sensitive to the balance of classes in the training data (see Kubat 2017, p 194).

RC: Page 11 Figure 7. This figure is very confusing and not self-explanatory at all. Please give more information in its caption, relative to the number in parenthesis (neighbor order I suppose), state that they are all or a few of the final parameters used.

AR: In P 11 Figure 7 the following caption will be added: "Parameters selected to minimize the classification error by the QDA method. From left to right: Amplitude of the homogeneity for the 2nd neighbour, integral of the contrast for the 18th neighbour, amplitude of the contrast for the 4th neighbour, integral of the correlation of the 8th neighbour and symmetry of the homogeneity for the 2nd neighbour. " will replace the initial "Figure 7: Texture analysis parameters selected to minimize the classification error of the training ensemble by the QDA method.".

RC: Page 12 Table 5. Change "eye-made" to a better term, like visual classification or similar.

AR: In P 12 Table 5 the word “visual” will replace the initial “eye-made”.

RC: Page 12 line 286. It is not clear if streaks were also detected during daytime, since the previous definition of the training set (line 162) says only night-time, but figure 9 says the opposite. Same for rolls and thermals. In summary, the constraint you talk about (day-time rolls and thermals, night-time streaks) does concern only the training set definition?

AR: Up until Figure 7 of the manuscript we showed the results for the training ensemble, where we only considered rolls and thermals during the day and streaks during the night. So the classification error of the algorithm in Figure 7 refers to the training ensemble. The algorithm was able to classify our training ensemble successfully for approximately 91% for the texture analysis parameters of Figure 7. Then we use these parameters to classify all the 4577 scans and the results are presented in Figures 8 and 9 of the manuscript, thus we detect streaks during the day and thermals and rolls during the night.

RC: Page 12 line 292. If I am correct, you tried to explain thermals during night, not others during days. Only the last word in the sentence, “reverse”, explains this. Moreover, can you elaborate more on what is the reason behind the erroneous classification of thermal as “others”? During stable conditions turbulent eddies are smaller, structures also show smaller length-scales. However, mean wind can show slight differences with no directional preference, and they can look like thermals (see Shah and Bou-Zeid, 2014).

AR: The category “others” includes the patterns that cannot be classified to one of the three turbulent structures type. It is possible to have a not symmetrical wind field, thus a bad case, during the day as well. In fact, 10 bad cases of the training ensemble for the category “others” occurred during the day. The physics behind the misclassification are very interesting and we are thankful for this insight. However, in our case the misclassification is linked to the shape of the patterns. It is possible that another texture

analysis parameter could improve the distinction between these two types.

RC: Page 13 line 295. Stable cases during night show buoyant forces opposing vertical momentum flux and turbulence generation. Mechanical turbulence does die out under stable conditions. Mechanical turbulence destruction by buoyancy is the dominant mechanism, not the opposite.

AR: This is true. We should probably have added in the text that we observe wind shear higher than 2 s-1 near the surface for at least 20 of the 62 days under study. This is the reason we expected the high number of occurrences for streaks during the night.

RC: Page 13 line 314. So thermals are not turbulent?. Why do you separate rolls and steaks form thermals? Does it has to do with pollution transport or something similar?

AR: Rolls and streaks are the focus on many boundary layer studies. In the specific sentence, we wanted to emphasize the regularity of observing coherent structures over Paris during the period of our study. Moreover, in the study we are currently working on with regards to the physics of the structures, the transition between the structures for particular cases (e.g. low level jets, cloud streets etc.) will be one of the focal points.

[Figure]

E - HAUTEURS
Zones Urbaines
Hauteur plafond en mètre

Territoires couverts par les fuseaux de protection du site de Paris
Zone Naturelle et forestière
Echelle : 1/25 000ème
0
500
1 000
1 500 M
PLU modifié les 6 et 7 février 2012
1er
2ème
3ème
4ème
5ème
6ème
7ème
8ème
9ème
10ème
11ème
12ème
13ème
14ème
15ème
16ème
17ème
18ème
19ème
20ème
Plan de Sauvegarde et de Mise en Valeur du 7ème arrondissement
Plan de Sauvegarde et de Mise en Valeur du Marais

**Fig. 1.**

[Figure]

**Fig. 2.**

[Figure]

**Fig. 3.**
[Figure]

-1
-2
-3
-4
-5
-5 -4 -3 -2 -1 0 1 2 3 4 5
Distance (km)
1
0.5
0
-0.5
-1
Turbulent radial wind speed (m/s)

Fig. 4.

[Figure]

Fig. 5.

[Figure]

Cosine fit at 11:38:50
observations
Radial wind speed (m/s)
1
0.5
0
-0.5
-1
-1.5

Azimuth angle

**Fig. 6.**

[Figure]

| | 1 | 2 | 3 | 4 | 5 | 6 | 7 | 8 |
|---|---|---|---|---|---|---|---|---|
| 1 | 69 | 0 | 0 | 0 | 0 | 0 | 0 | 0 |
| 2 | 0 | 69 | 0 | 0 | 0 | 0 | 0 | 0 |
| 3 | 0 | 0 | 69 | 0 | 0 | 0 | 0 | 0 |
| 4 | 0 | 0 | 0 | 69 | 0 | 0 | 0 | 0 |
| 5 | 0 | 0 | 0 | 0 | 69 | 0 | 0 | 0 |
| 6 | 0 | 0 | 0 | 0 | 0 | 69 | 0 | 0 |
| 7 | 0 | 0 | 0 | 0 | 0 | 0 | 69 | 0 |
| 8 | 0 | 0 | 0 | 0 | 0 | 0 | 0 | 69 |

**Fig. 7.**

---

## Author Response (AR1)

**Authors' response to the first referee**

1) Referee Comment: 1. In the manuscript a cosine fit is applied to wind lidar data acquired during VAD scans in order to estimate the mean wind speed and direction. Such a data analysis method requires the assumption of horizontal homogeneity

- 5 of the wind vector, as also stated by the authors. However, I am wondering to which extent this homogeneity is expected over the urban landscape of Paris. I think that a discussion on the terrain heterogeneity is missing from the article. In this direction, I think that it would be constructive to add an elevation map of the area over which the scanning wind lidar was acquiring measurements. This would provide an insight to which extent the observed spatial variations of the wind are related to temporal fluctuations of the wind and/or due to changes of the terrain elevation.
- 10 1) Authors' Response: 1. The lidar scans were covering the urban area of Paris as it can be seen in Figure 1b of the manuscript. In Paris there is a height limit for the buildings. Please see Figure 1 of the supplementary materials with the height limits of the buildings in Paris from the official Planning Department of the city of Paris, 2006. The Jussieu site is highlighted by the green text symbol. The buildings in the centre of Paris range between 25 and 37 m and rarely reach or exceed 50 m. So despite being an urban area, it is rather homogeneous. We have added the elevation map of the area in P5 as Figure 2 in the manuscript.
- 15 In Figure supplementary materials you can also find the map with the altitude of the beam, see Figure 2 of the supplementary materials.

2) RC: In this context and regarding the data analysis: a. A statistical parameter is required to specify the representativeness of the fit used in the VAD scan. In the manuscript it is stated that the RMSE values of the estimated fit have been calculated but they are not stated in the document. b. Did the authors perform a quality check of the acquired data? Do they apply any SNR filtering to the acquired radial wind speed prior the application of the data analysis?

2) AR: a. The RMSE and similar statistical parameters for the fit quality evaluation incorporate in their values the difference between the radial wind speed and the fitted function which is the parameter we are interested in. Therefore, it is complicated to use such a parameter to characterize the quality of the fit. What we did instead, it was to select the bad cases based on the

- 25 symmetry of the radial wind field. An example of a bad case is showcased in Figure 4a of the manuscript, when the radial wind field is not symmetric and as a result the radial wind speed shows a more chaotic behaviour when plotted against the azimuth angle (Figure 4, manuscript). On the contrary the good cases were selected for symmetric radial wind fields as the one presented in Figure 3 of the manuscript.
- b. We forgot to include this information. We considered only the radial wind speed values for which the carrier-to-noise ratio
   (CNR) is higher than -27 dB. The values where CNR< -27 dB were filtered out since the radial wind speed had anomalously high values, 2 times or higher compared to the rest of the radial wind observations. In P5 L149-151, the following sentence has been added "For this study, the radial wind speed values for which the carrier-to-noise ratio is lower than -27dB (CNR<-27dB) were disregarded since they were anomalously high, exceeding the values of the rest of the radial wind field by two times or more.".</li>

35

3) RC: 2. The subtraction of the mean wind speed from the radial wind speeds does not compensate the fact that the individual measurements along the scanning pattern are the result of the projection of the instantaneous wind vector to the line-of-sight of the wind lidar. Therefore, the term turbulent wind field could lead to a misinterpretation. The authors should clearly state that they measure the high frequency fluctuations of the radial wind speed.

40 3) AR: It is true that using the word turbulent can be misleading since we do not observe very small scales. We will make clear in the manuscript that what we observe are medium-to-large fluctuations and coherent structures (mlf-cs). However, we will state that these are associated to a turbulent atmosphere.

 4) RC: Specific Comments: P2 Line 28. The reference of Roth 2007 is a review of the atmospheric turbulence studies over urban landscapes and it doesn't directly discuss how the pollution concentration in urban environment is dependent on the weather and on the turbulence. A reformulation of this sentence is suggested for a clarification.

4) AR: The first paragraph will be removed along with the references. We will instead emphasize the effect of the turbulent structures in the pollutants' dispersion. In P1 L31-46, the following text has been added: "Several studies have been carried out to examine the effect of the coherent turbulent structures in the dispersion of pollutants by utilizing boundary layer simulations. The results of these studies indicate that the coherent structures can play a significant role in the pollutants' concentrations (Aouizerats et al., 2011; Soldati, 2005). Furthermore, (Sandeepan et al., (2013) have demonstrated via

simulations that the pollutants' concentrations can alternate from low to high during coherent structures events. It is therefore important to be able to identify structures in the atmosphere and observe them in an efficient and consistent way.".

55 5) RC: P3 Line74. Why is it relevant if the campaigns are short-term or long-term? And what is the time scale of these two types of campaigns that it is relevant for the topic of this study?

5) AR: We used an example of a short term study to show that so far lidars have been used to observe coherent structures for short periods since the analysis of a large dataset is very time-consuming. On the other hand, the example of a long-term study is mentioned to show that it provides us only point measurements instead of the whole wind filed over an area. Thus we wanted

60 to highlight the benefits of our study. In order to clarify the limitations of the long-term study, we have included the following sentence in P3 L82-83 "However, their study is limited to point measurements instead of a larger wind field that it is possible to observe via a lidar."

6) RC: P3 Line95. I am not sure that the two references of Kumer 2014 and Veselovskii et al. 2016 are the appropriate to be 65 used here. In none of them there is an analytical explanation of the wind lidar instrument as it is stated. I suggest replacing them with more relevant references. An example could be the work of: Cariou, J. P., R. Parmentier, M. Valla, L. Sauvage, I. Antoniou, and M. Courtney. "An innovative and autonomous 1.5 µm Coherent lidar for PBL wind profiling." In Proceedings of 14th Coherent Laser Radar Conference. 2007.

6) AR: Cariou et al, 2007 has replaced the references of Kumar and Veselovskii in P4 L109 as it is more appropriate.

70

75

7) RC: P3 Line 101. It is stated that "The duration of each scan was 3 minutes which is sufficiently short for the observation of structures". Could the authors elaborate more on this statement?

7) AR: In P4 line 115, the word "fast" will replace the initial "short". Furthermore, the sentence "The duration of each scan was 3 minutes which is sufficiently short for the observation of structures" has been rephrased to "The duration of each scan was 3 minutes which is sufficiently fast for the observation of coherent structures, as their lifespan is several minutes."

8) RC: P4 Line 102. The authors state that the maximum range of the scans reached 5 km. However, in none of the figures that they have included in the manuscript the range ever reaches 5 km. The most common range in their data is between 2 -2.5 km. Can the authors explain why is this happening?

80 8) AR: We have used the CNR < -27 dB filtering. Please see also the reply 2b.

9) RC: P4 Line111. Table 2. A list of scanning patterns is included here that are not used in this study. In addition, the purpose of each scanning pattern is included without explaining the reasoning for their selection. I would suggest to either remove this table. If the authors wish to keep it then I suggest that they should elaborate more in the text about it.

- 85 9) AR: We acknowledge that the presentation of the entire scanning sequence can be confusing for the reader, therefore the Table 2 in P5 has been modified to include only the information for the PPI and the DBS scans. We have also removed the 90° elevation angle from the table as it was not part of the DBS scanning method. The latter scanning method is also important for this study as the DBS scans revealed the high wind shear cases during night. The sentence in P4 L111 "Table 2 showcases the implemented scanning methods during the VEGILOT campaign" has replaced the initial "Table 2 showcases the scanning
- 90 sequence as it was implemented during the VEGILOT campaign.". Additionally, a brief description of the DBS scan will be added to the text. In particular, the following sentence has been added in P4 line 122-125: "It was also important for this study to retrieve observations regarding the vertical wind shear. For this purpose, the Doppler beam swinging (DBS) scanning method was implemented. This method was consisted of four line of sight beams at azimuth angles of 0°, 90°, 180° and 270° with an elevation angle of  $75^{\circ}$  and it was applied twice. The duration of the four direction beams emission was approximately 95
- 15 seconds.".

10) RC: P4 Line120. The "offset" refers to the vertical wind speed component? What does it mean that  $\alpha$  is much smaller than b? Is this a common observation over the whole scanning plan? And what kind of  $\alpha$  values does the application of the model result in?

100 10) AR: The offset is indeed associated to the vertical wind speed. The radial wind speed analysed in its' wind components for a line of sight beam with azimuth angle  $\theta$  and elevation angle  $\phi$  will be:

 $Vr = u \cos\theta \sin\phi + v \sin\theta \sin\phi + w\cos\phi$ , where u is the longitudinal, v is the transverse

and w is the vertical wind component respectively. The offset is associated to the parameter wcos (Thubois et al. 2018, Study

- of the configurations and scanning strategies of Doppler Lidars for providing wind and aerosol/cloud profiles). Since the 105 elevation angle of the PPI scans was 1° and not 0° we included this parameter in the fit function. However, it is much smaller than the b component which is associated to the horizontal wind. So the horizontal wind is still dominant. This is a common observation over the whole scanning plan. The value of the offset is around 10 times smaller than the amplitude and ranges according to the value of the amplitude from 0.05 to 0.5 m/s.
- 110 11) RC: P6. Line156-157. It is not clear how the unaligned thermals are dependent on the increased solar radiation measurements.

AR: The increased solar radiation measurements result in surface heating which we typically observe during fair weather cumuli conditions. The phrase in P7 L195 has been modified to "Regarding unaligned thermals, solar radiation measurements ....., signifying fair cumuli weather conditions."

115

120

125

12) RC: P6. Line 163. How do the authors estimate the value of the wind shear? Also, shouldn't the units of the wind shear be seconds to the power of -1?

12) AR: The DBS observations provided us the zonal u and meridional v winds. The horizontal wind was computed from these components from the formula: V hor =  $\sqrt{u^2 + v^2}$ . The formula has been included in the text as Equation 3 in P7. The wind shear was estimated from the vertical profile of V\_hor by subtracting the local minima from the local maxima above it, near the surface. The units have been corrected to s-1.

13) RC: P6. Line 166. The authors in P5. Line 133 state that the VAD method could not be applied in the data acquired during the night, especially at those occasions where the mean wind speed was less than 2 m/s. Does this mean that only cases with mean wind speed higher than 2 m/s were selected?

13) AR: The VAD method can be applied regardless of the mean wind speed values as long as the radial wind speed field is symmetric. It is true however that when the mean wind speed is lower than 2 m/s then the radial wind field has the risk to not be symmetric. For the training ensemble, regarding streaks, rolls and thermals only cases with moderate winds were selected (5-8 m/s) whereas for the category others cases with weak winds below 2 m/s were included. We are currently preparing a separate study with the physics of the structures based on the whole classification from this methodology. The results show

- 130 that we observe winds below 2 m/s mostly during the others category. More particularly the results are: 7 out of the 1145 streaks cases, 0 out of 420 rolls cases, 67 out of the 900 thermals cases and 670 out of the 2112 others cases.
- 14) RC: P6. Line 167. How well was the mean direction and speed estimated through the VAD method in the selected cases? 135 I suggest adding a statistical parameter that describes the representativeness of the applied fit (equation 1) to the measured data (e.g. RMSE).

14) AR: As we previously stated statistical parameters as the RMSE are not appropriate for the evaluation of the fit since the interesting parameter for us is the difference between the observation and the fit. We instead selected the training ensemble based on how well symmetric or not was the radial wind field and by plotting random rings for each scan to confirm good fits

140 as displayed in Figure 3 or a bad fit as in Figure 4 of the manuscript.

15) RC: P7. Line 170. It is difficult to understand the scale of the Modis colour images. Would it be possible to either add a scale or mark on the images the scanning area?

15) AR: In P8 in Figures 5d and 5e, the scanning area has been marked on the MODIS images.

16) RC: P7. Line 176. I recommend describing very shortly the texture analysis especially in the context of remote sensing. It would be useful to add any references in the introduction regarding the previous applications of this type of analysis to remote sensing data.

16) AR: In P3 L86-88, we have added the following brief description: "Texture analysis is an effective way to evaluate the distribution of the values within an image (Castellano et al., 2004). It is widely used in various scientific fields in order to classify images, covering meteorology (Alparone et al., 1990), medical studies (Holli et al., 2010) and forestry (Kayitakire et al., 2006).".

**17) RC: P8. Line 181-182. What is the logic behind the selection of this particular values for defining the contrast?**

- 155 17) AR: We wanted to enhance the contrast of the structures. For this reason, we had to select the bins in such a way that the difference between positive and negative values will be more apparent. We have tried more configurations. The selection of only 2 bins (one positive, one negative), led to a less successful classification of the different types of structures as the 2 bins give a 2 by 2 co-occurrence matrix. The size of the co-occurrence matrix is important in this case, since from the Equations 2, 3 and 4 of the manuscript, it is evident that the texture analysis parameters also depend on the distance between the bins *i,j*.
- 160 The classification error in this case was around 18%. The selection of the 4 bins (one bin including all the negative values below -0.5 m/s, one bin between -0.5 m/s and 0, one bin between 0 and +0.5 m/s and one bin including all the positive values above +0.5 m/s) did not really improve the results with the error remaining around 18%. The selection of 8 bins reduced the error significantly. The selection of one bin including all the negative values below -0.5 m/s, six bins equally distributed between -0.5 m/s and +0.5 m/
- 165 between the positive and negative values while keeping the distance between the bins *i,j*. We did not try to select more than 8 bins because we believe that it will not be useful to increase the number of bins near 0. It is in our future plans however to automatize this part as well. We would like to include an algorithm in our method in order to find the optimal selection of bins that minimizes the classification error.
- 170 18) RC: P10. Line 243. The authors state that 60 cases of "other" patterns are used during the supervised machine learning step. They argue that this is necessary because "it is expected to be the dominant category in the classification". I am not sure that I understand what it is meant with this statement. I suggest having this part a bit more explained. Furthermore, how is the mean wind speed and direction estimated in these cases? As it is stated the VAD was not successfully applied to this data.
- 18) AR: When we were analysing the results, we observed that the chaotic type of patterns (see Figure 3 of the supplementary materials) was the most common type. The algorithm can be sensitive to an unbalanced training ensemble. It is preferable to select a training ensemble based on the expected results (Kubat, 2017 p.194). Therefore, even for the bad cases, the VAD method was selected for the estimation of the mean wind speed and wind direction. In Figure 4 of the supplementary materials, a fitted function for a non-symmetric field is displayed. We can still obtain the mean wind direction from this figure, but the patterns will have the chaotic look as in Figure 3 of the supplementary materials.
- 180

19) RC: P10. Line 225. How do the authors explain the change in the slope of the homogeneity curve that is observed for absolute azimuth angles larger than 45 degrees?

19) AR: The angle represents also the distance between two grid points. For 45° angles or above, the distance between two grid points are n rows away whereas below 45° they are n-1, n-2 etc. We have prepared an illustration of an ideal case, Figure 5 of the supplementary materials. We hope that it is clear. Keep in mind that the Figure 7 of the manuscript refers to the third neighbour which is equivalent to 150 m distance between the grid points.

20) RC: P10. Line 240. Could the authors add a reference to the literature describing the "supervised machine learning methodology"?

190 20) AR: In P11 L281, the following references have been added:

"Bonamente, M. Statistics and analysis of scientific data, Springer, 2017, 318 p., DOI:10.1007/978-1-4939-6572-4;

Gareth James, Daniela Witten, Trevor Hastie, Robert Tibshirani, An Introduction to Statistical Learning with Applications in R, Springer Texts in Statistics, 2013, 426 p, DOI: 10.1007/978-1-4614-7138-7

Kubat M. An Introduction to Machine Learning, Springer, 2017, https://doi.org/10.1007/978-3-319-63913-0"

200

21) RC: P11. Line 265. How do the authors physically explain this result? A low RMSE in the cosine fit, couldn't also mean that the mean wind speed and direction are not estimated correctly?

21) AR: This result show that according to the algorithm, the different shapes of the patterns in the radial turbulent wind field are more significant for the classification of the structures than the physical parameters. The wind values range from 1 m/s to 14 m/s. Therefore, a higher RMSE does not necessarily mean a worse fit, but it may be caused due to this scale difference.

22) RC: P12. Line 291. It is stated that "there were scarcely any rolls cases observed at night". However, in the sentence 288 it is written that for the classification of this study the authors "consider only thermals and rolls during daytime". To my understanding there is an inconsistency between the two statements. Can this be explained better?

- 205 22) AR: For the training ensemble only thermals and rolls during daytime were selected. However, the classification of all the data was made based on the five texture analysis parameters displayed in Figure 8 of the manuscript. As time is not included in the classifiers, the algorithm can classify the patterns at any time of the day but still few cases of rolls and thermals were classified during the night. This result is an indication that the classification is working as intended. In a study of the physical properties of the structures, rolls and thermals during night will be excluded.
- 210

23) RC: P13. Line 312. In the conclusions the author state that time, wind speed and the cosine fit RMSE of the VAD method were not selected by the algorithm for the classification. However, in the results presented in figure 9 there is a time dependency in the detection of certain patterns (e.g. thermals and rolls). Could the authors comment why the inclusion of the time as a classification parameter would not improve further their results?

- 215 23) AR: The algorithm finds the best combination of parameters that minimize the classification error and time was not one of the five parameters as it can be seen in Figure 8 of the manuscript. By including the time as a parameter the classification error will not be reduced.
- 24) RC: P13. Line 318. Given the fact that one PPI scan lasts for 3 minutes and occurs every 18 means, can the authors explain how does the acquired data set contribute to the comprehension of the development of coherent structures?

24) AR: The lifespan of streaks can be several tens of minutes and for rolls it can be hours. Even with this time gap between the observations we believe that it is still interesting to study the transitions between the different types of structures. For example, we have vertical lidar observations between the PPI scans and we can study the development of the atmospheric boundary layer height with regards to the type of the structures.

**225**

25) RC: Technical Corrections: General: There is a small inconsistency on the way that figures are referred to. Sometimes they are used parenthesis after the number of the Figure to denote a subfigure and sometimes are not.

25) AR: All the subfigures have been corrected in order to keep consistent naming. The number of the subfigure is now followed by the corresponding letter e.g. "Figure 1a".

**230**

26) RC: P2 Line 34. I think that the statement "Futhermore, ...." is a self-evident. I would suggest removing it. Also, I would suggest to add the reference of (Hussain, 1983) at the end of the previous sentence.

26) AR: In P2 line 34, the sentence: "Furthermore ..... time-averaged statistics calculations." has been removed. The reference of (Hussain, 1983) now refers to the sentence in P2 L28-29: "The principal aspect that determines a coherent structure is the maintenance of the phase-averaged vorticity of the turbulent fluid mass over the spatial extend of the flow structure.".

235

27) RC: P2 Line 37. "and the lower"

27) AR: In P2 line 42 "and the lower" has replaced "and lower".

240 28) RC: P3 Line 72. I suggest reformulating this sentence. It is not clear to the reader how is the two-dimensional autocorrelation function was used. Also, I suggest changing the sentence "the observation of the scans by eye" to "visual observation of the scans".

28) AR: In P3 L75-77, the sentence "They combined quantitative characteristics of the coherence such as the integral scales and the anisotropy coefficients, obtained by a two-dimensional autocorrelation algorithm, with the visual observation of the

245 scans." has replaced the initial "They combined a two-dimensional autocorrelation function with the observation of the scans by eye".

29) RC: P3 Line76. Change "month" to "months"

29) AR: In P3 line 81 "months" has replaced "month".

**250**

30) RC: P3 Line83. The "Section 0" should change to "Section 3"

30) AR: In P3 line 96 "Section 3" has replaced "Section 0".

31) RC: P3 Line86. The word "two" is used as a noun modifier and therefore the word month should be in singular form.

255 31) AR: In P3 L100, the phrase "A two-month measurement campaign" has replaced the initial "A two-months campaign".

32) RC: P3 Line96. I suggest to change the text "The lidar is sensitive only to the" to "A wind lidar is measuring the"

32) AR: In P4 line 110, the phrase "A wind lidar is measuring the" has replaced the initial "The lidar is sensitive only to the".

260 33) RC: P3 Line100. Change the "for a" to "with a".

33) AR: In P4 L113 "with a" has replaced the initial "for a".

34) RC: P4 Line103. I think that it is more grammatically correct to either use the past tense or the passive form of the "rise" verb.

265 34) AR: In P4 L121 "the beam rise" has been rephrased to "the beam was risen".

35) RC: P5 Line132. The authors state that due to the surface heterogeneity the VAD method can be applied in some cases. A surface heterogeneity will introduce an error in the VAD method regardless the wind speed.

35) AR: We have probably phrased this in a wrong way. As it can be seen in Figure 2 of the supplementary materials, there are some hills in the limits of the scanning range but with low elevation. We have not study what could be the orographic effect in the wind speed as this would require model simulations. In P6 L168-170, the following sentence had been added: "Troude et al. (2002) and Lemonsu and Masson (2002) have performed numerical weather simulations in the area of Paris and have observed that during low wind conditions (below 3 m/s) the orographic effect and the urban heat island effect could be the main drivers for the local wind speed.".

**275**

- 36) RC: P5 Line132. I suggest that the "Jussie site" is changed to "The Jussie site".
- 36) AR: In P6 L167 "The Jussieu site" has replaced "Jussie site".

37) RC: P5 Line138. Figure 3 caption. Change the "a case" to "A case", also a add a tab space between (a) and "Radial".

280 37) AR: In P6 L174 "A case" has replaced "a case" and a tab space has been inserted between (a) and "Radial" in the caption of Figure 4.

38) RC: P6. Line149. "Sec" should be replaced by "Section".

38) AR: In P7 line 149 "Section" has replaced "Sec".

285

39) RC: P6. Line 164. I suggest to re-write the sentence "For many cases, the wind shear was accompanied by turbulent streaks pattern" and specify for which particular wind shear values were streaks detected.

40) RC: P6. Line165. The part of the sentence "so or the training ensemble" should be rewritten.

39-40) AR: In P7 L204, the sentence "For the training ensemble, only night cases when streaks patterns (Figure 5c) were 290 accompanied by wind shear higher than 2 s-1 were selected." has replaced the initial "For many cases, the wind shear was accompanied by turbulent streaks patterns (Figure 4c) so or the training ensemble, only night cases of streaks were selected to ensure that wind shear was the primary factor for the generation of turbulence.".

41) RC: P10. Line 230. "fore" should be changed to "for".

295 41) AR: In P11 L271 the word "for" has replaced the initial "fore".

42) RC: P11. Line251 – 253. The authors state the higher number of dimensions relative to the number of patterns lead to the "curse of the dimensionality problem". I suggest to re-write it by using appropriate scientific statements.

42) AR: The "curse of the dimensionality problem" is a scientific term, which is common in statistics/data science domain. 300 We think that it is relevant here.

43) RC: P11. Line 259. Correct the "Section 0".

43) AR: "Section 0" has been removed from that part of the text.

305 44) RC: P12. Table 5 caption: I suggest changing the "eve-made" to "visual".

44) AR: In P13 the word "visual" has replaced the initial "eye-made" in the caption of Table 5.

45) RC: P12. Line 290. The coherent structures don't have a preference. They are formed under favourable atmospheric conditions. I suggest commenting the result of Figure 9 on that basis.

- 310 45) AR: This was undoubtedly a bad way to phrase it. In P14 L352-356, we made the following changes in the text: "It is evident that despite time was not one of the selected classifiers, the number of occurrences of the structures show a distribution that can be associated to the atmospheric conditions. More particularly, rolls and thermals were mainly classified during day. This result is noteworthy as these structures are linked to a well-developed atmospheric boundary layer during day. On the contrary, there were scarcely any rolls cases observed at night and a few unaligned thermals were classified at night." on the
- 315 initial "It is evident that despite time was a much less significant classifier compared to the curves parameters, the structures show a time preference. There were scarcely any rolls cases observed at night, though a few unaligned thermals were classified at night."

46) RC: P15. Line 386. "Lemone" should be changed to "LeMone". Also, necessary information of the article (e.g. journal) 320 is missing.

46) AR: In P118 L488, the name has been corrected and the missing information has been added. The complete reference now is: "LeMone, M., 1972. The structure and dynamics of the horizontal roll vortices in the planetary boundary layer. J. Atmos. Sci. 30, 1077–1091. https://doi.org/10.1175/1520-0469(1973)030<1077:tsadoh>2.0.co;2"

**325 Authors' response to the second referee**

47) RC: As I understand form the first paragraph in the introduction, comprehension of the flow physics it is important for monitoring atmospheric pollution. However, the physics identified here, in the form of coherent structures, are not related with bad pollution conditions, since the latter might fall in the "Other" category, and their physics are not clear from the study.

330

47) Author Response: We recognize that the first paragraph can be misleading and therefore it has been removed. We focus only on the effect of turbulent structures in the pollutants' dispersion. See also the response (4) to the first author.

48) *RC*: The motivation of the study should be stressed form the beginning. Only when we arrive to the conclusions we can read some of the potential application of this approach. (line 316 of the text).

48) AR: We want to show that it is possible to identify and classify these structures based solely on the patterns from the fluctuations of the radial wind speed data by combining texture analysis and supervised machine learning. For this reason, we have added the following text in the Introduction P3 L83-93 last paragraph: "This study aims to identify the medium-to-large fluctuations and coherent structures (mlf-cs) on single Doppler lidar horizontal scans and develop an automatic classification

- 340 process based on the combination of texture analysis and a supervised machine learning technique, namely the Quadratic Discriminate Analysis (QDA), in order to handle large datasets. Texture analysis is an effective way to evaluate the distribution of the values within an image (Castellano et al., 2004). It is widely used in various scientific fields in order to classify images, covering meteorology (Alparone et al., 1990), medical studies (Holli et al., 2010) and forestry (Kayitakire et al., 2006). There is a lack of long-term studies of structures based on lidar observations and the aforementioned automatic classification process
- 345 can stimulate the interest in this research field. More particularly, it could facilitate the statistical analysis of the physical parameters of the structures, e. g. the structure size as a function of the planetary boundary layer (PBL) height. Furthermore, it will enable us to study the transitions between structures and how these are associated to the atmospheric conditions. Finally, the impact of the structures on pollutants' concentrations could be examined for long-term studies under stable and unstable conditions."
- 350

49) RC: In general, the term turbulence and turbulence fields are used frequently in the text, but the range gate resolution of the lidar scanner is 50m at a height of 75m above ground level. Turbulence and its most energetic eddies might fall within this length scale. Almost all turbulence fluctuations are filtered out by the lidar due to spatial averaging. What we can clearly see from lidar observations is medium-to-large scale fluctuations and coherent structures rather than turbulence.

49) AR: We have replaced the term turbulence to the suggested medium-to-large fluctuations and coherent structures (mlf-cs) as we do not observe small scale turbulence and the reader can be confused. Nonetheless, we have stated that these structures are associated to a turbulent atmosphere.

50) RC: I miss a paragraph describing how data quality was assured.

50) AR: Unfortunately, during the VEGILOT campaign there was no other wind data measurements for comparison. The closest weather station with available data for the same period, as our study, is located in Montsouris, 20 km away from the centre of Paris.

**51) RC: Did you use a CNR/SNR threshold for filtering?**

51) AR: We used the CNR filtering. The radial wind speed values below -27 dB were anomalously high and therefore excluded from the computations. See also response (2b) to the first author.

**52) RC: What was the data recovery rate during the two months of measurements?**

52) AR: The lidar was taking measurements continuously for the two-month period for the scanning sequence presented inTable 2 of the manuscript. There was no pausing of the measurements during this period.

53) RC: It is not clear from the text how the radial wind speed fluctuations are calculated. What does "stronger radial wind speed" mean? A larger absolute value? It seems to me that the sign might come from the combination of u and  $\cos(\theta)$ . One suggestion is to put this definition as equations.

- 375 53) AR: The radial wind field has values with a positive sign when the wind moves away from the lidar and negative sign when it moves toward the lidar. In order to be consistent throughout the field when we study the fluctuations, we have to make a sign convention that guarantees for example that the radial wind speed with a value of 6 m/s and the radial wind speed with a value of -6 m/s will be part of the same pattern. Therefore, we compared the absolute values and we derive the sign convention of Figure 2b. We have add the following formula in the manuscript as Equation 2 P5 L162:  $u'_r = |u_r(\theta)| |f(\theta)|$ , where f is
- 380 the fitted function for the corresponding azimuth angle  $\theta$ .

54) RC: Since the wind direction is obtained from equation (1) it would be possible to work with the streamwise component u instead of the radial wind speed ur. It is not clear from the results, what is the relative importance of non-distinguishable structures, bad fittings, and bad data (low CNR/SNR signals) in the "Others" category. The authors give some information about what it seems to be the reason of one group of cases to belong to this category (bad fitting of cosine function), but no threshold on this fitting error is given.

54) AR: We did not perform an extensive analysis of the data for the "Others" category. This category was selected exclusively to separate the not interesting patterns. The bad cases were not selected based on the quality of the fit but rather in the symmetry of the whole radial wind field. For the category "others" in the training ensemble, we selected 53 bad cases with an

- 390 asymmetrical radial wind field and 7 cases of a symmetrical field where the structures were non-distinguishable. In Figure 3a of the manuscript, it is apparent that the wind field is not symmetrical and that the VAD method would not be applied efficiently. Nevertheless, it can still be applied as you can see in Figure 4 of the supplementary materials (it is the same with Figure 4b of the paper along with a cosine fitted function). Since we are interested in the fluctuations (difference between the observation and the fitted curve) it is very difficult to characterize the quality of the fit by using a parameter such as the RMSE
- 395 that contains these fluctuations. We are currently working on a study for the physical parameters of the structures based on the classification of this study. We found that for 670 out of the 2112 others cases the mean wind speed is lower than 2 m/s. On the other hand, for the streaks it is only 7 out of the 1145 cases, for rolls 0 out of 420 cases and for thermals 67 out of the 900 cases. This is very interesting as the mean wind speed was not one of the classifiers.
- 400 55) RC: I miss more elaboration in the description of the texture parameters used for classification, namely, why they might be relevant and if they were relevant in the end. The feature selection process is also not very clear. Cross validation is well known and well explained, but the text explaining the outcome as well as the figure used in that regard are confusing. A brief description of the machine learning technique used could be useful for clarity.
- 55) AR: The description of the texture parameters has been added in the manuscript, in particular in P10 L251-255: 405 "Correlation indicates the existence of linear structures in the image, with high values associated to a large amount of linear structure in the image. Contrast reveals the local variations in an image, where a large amount of variations leads to high values. Homogeneity is self-explanatory and the high values represent a homogeneous image. Finally, energy measures the uniformity of an image with the highest values corresponding to constant or periodic forms (Haralick et al., 1973; Yang et al., 2012).".
- 410 These texture parameters are frequently used in patterns based image classification. As we mention in our manuscript we selected these four parameters inspired by the study of Srivastava et al. (2018). They used the same parameters to distinguish stripes among others patterns. By plotting the texture parameters against the azimuth angle  $\varphi$  (angle of the comparing neighbour points), we observed a prominent peak for the elongated patters for  $\varphi$  = mean wind direction. The relevant parameters are showcased in Figure 8 of the manuscript.
- 415 Regarding the feature selection process by the algorithm the following text has been added in P10-11 L287-293: "The greedy algorithm of stepwise forward selection was used in the article, which is the standard and frequently used method of reduction of the feature space. As indicated in (Sokolov et al., 2020), it can be formulated as follows. The features are divided into two groups accepted in the classification model and remaining, for which an estimate of the possibility of acceptance into the model is checked. Features from the set of "remains" are consecutively added to the model and corresponding estimations of
- 420 the classification error are calculated. From the received set of errors, the minimum is chosen and compared with the error of the previous model. If a significant reduction of the error occurred, then the corresponding feature is accepted into the model, if not then the process stops."

For the specific machine learning technique, we utilized, we have included a brief description in P11 L283-286: "Quadratic discriminant analysis or normal Bayesian classification (Hastie et al., 2009) is the parametric approach implying that

425 probability density functions (PDF) belong to the family of normal distributions. It is a classical algorithm of the supervised machine learning, based on the principle of maximum likelihood. The general idea is to estimate the PDF for each class, and then select the most probable class (Kubat, 2017)."

56) RC: Conclusions section. In my opinion, this section should be read in a positive rather than negative way. Example: it
 should focus on the relevant parameters discovered (which need a bit more explanation in the corresponding section) rather
 than the ones excluded by the study. The sections describing the methodology used-which need some improvement-are already
 clear, and no repetition is needed. Same with the results highlighted.

56) AR: It is very important for us to stress the positives of this study and therefore we have modified the conclusions to comment on the relevant parameters. In P15 L376-384, the following text has been added: "More particularly, these parameters were the amplitude of the 2nd-neighbour homogeneity curve and the amplitude of the 4th-neighbour contrast curve which were associated to the prominent peaks of the elongated patterns (streaks, rolls). Furthermore, the integral of the 18th-neighbour contrast curve and the integral of the 8th-neighbour correlation curve which could distinguish, for example, chaotic patterns ("others") with high contrast and lower values of correlation between neighbour points compared to an enclosed homogeneous (thermals). Finally, the symmetry of the 2nd-neigbour homogeneity curve revealed the importance to align the

- 440 mlf-cs fields to the mean wind direction. Another striking outcome of the QDA classification was the variety of the classifiers in terms of distance between the grid points. The 2nd-neighbour translates in a distance between two grid points equivalent to 100 m and for the 18th-neighbour to 900 m. This is essential for the classification between patterns with different sizes such as streaks and rolls."
- 445 57) RC: 2 Specific comments Page 1 Line 12. Change "manually" to "visually".
  - 57) AR: In P1 L12 the word "visually" has replaced the initial "manually".

58) RC: Page 1 Line 15. Change "and installed" to "installed".

58) AR: In P1 L15 the word "and" has been removed with the phrase now reading "by a scanning Doppler lidar (LEOSPHERE 450 WLS100) installed".

59) Page 1 Line 16. It would be better to reword this sentence, maybe "The turbulent component of radial wind speed is estimated using...over 4577 scans.

59) In P1 L16-17 the sentence "The lidar recorded 4577 quasi-horizontal scans for which the turbulent component of the radial wind speed was determined using the velocity azimuth display method." has been rephrased to "The mlf-cs of the radial wind speed are estimated using the velocity azimuth display method over 4577 quasi-horizontal scans.".

60) RC: Page 1 Line 18-21. I am not sure what the sentence describing the training set adds to the abstract if not combined with the next one. It is better to state directly the unsupervised algorithm used instead of using parenthesis. It might be better to rephrase this in a more concise way.

60) AR: In P1 L19-23 the text: "The differences between the three types of structures ..... Using the 10-fold cross validation method, the classification error was estimated to be about 9.2% for the training ensemble and 3.3% in particular for streaks." has been rephrased to "The differences between the three types of structures ..... The algorithm was able to classify successfully about 91% of the cases based solely on the texture analysis parameters. The algorithm performed best for the streaks structures with a classification error equivalent to 3.3%."

465

460

61) RC: Page 1 Line 23. What are the remaining 20

61) AR: The remaining 20% are the unaligned thermals. We wanted to highlight the results only for the rolls and streaks as we find in literature that the majority of the studies focus on these two types of coherent structures.

62) RC: Page 2 Line 26. Change "step for" to "step towards".

62) AR: We have removed the first paragraph of the introduction and instead address only the impact of turbulent structures on pollutants' dispersion.

475 63) RC: Page 2 Line 32. A coherent structure is defined according to its phase-averaged rather than its instantaneous vorticity. I also suggest moving the Hussain 1983 reference to this sentence. A coherent structure needs to maintain its phase-averaged vorticity rather than its time-averaged vorticity or form.

63) AR: In P2 line 32 the adverb "instantaneously" was referring to the phase-averaged vorticity. In order to avoid confusion, the text in P2 L28-29: "The principal aspect that determines a coherent structure is the instantaneously space and phase correlated vorticity of the turbulent fluid mass over the spatial extend of the flow structure. Furthermore, a coherent structure must maintain its form for a time period sufficient for time-averaged statistics calculations (Hussain, 1983)." has been rephrased to "The principal aspect that determines a coherent structure is the maintenance of the phase-averaged vorticity of the turbulent fluid mass over the spatial extend of the flow structure.

485 64) RC: Page 2 Line 35. Please specify that this is the case of atmospheric flow. Other structures are observed at laboratory scale (also in the atmosphere but not so relevant for momentum or scalar fluxes), like hairpins, or hairpin trains. Include a reference to Hutchins and Marusic (2006) and Adrian (2007).

64) AR: We have included the following text in P2 L36-39: "The term coherent structures in the aforementioned studies refers exclusively in the atmospheric flow and it is the main focus in this study. This term is also encountered in studies at laboratory scale described as hairpins or packets (Adrian, 2007; Hutchins and Marusic, 2007), but these are out of the scope of this study.".

65) *RC*: Page 2 lines 37-44. Consider reordering the sentences here for a more fluent reading. Maybe starting the paragraph from sentence in line 41?

65) AR: We have rearranged the sentences in P2 L40-44. The first sentence of the paragraph is now: "Turbulent streaks are
 structures aligned with the horizontal wind with alternating stripes of stronger horizontal wind associated with a subsidence and stripes of weaker horizontal wind associated with an ascendance (Khanna and Brasseur, 1998).".

66) RC: Page 2 line 45. How is it that you identify rolls in the mixed layer, with sizes from few to dozen kilometers, with scans at surface layer height (75m) with spatial coverage of less than 2 kilometers?. Is this description coherent with what you are identifying?

66) AR: In the introduction we give general information about the rolls and we think it is important to address their scale. In P7 L192-193, we have added the following sentence: "It is important to note that we observed the occurring patterns near the surface, hence near the lower part of the rolls."

505 67) RC: Page 2 lines 57-65. Since you are using lidar instead of radars it would be better to shorten the scanning pattern description using some of the given references, since they have to do only with the history on scanning patterns. There are more recent references of this regarding lidars. Cariou (2007) and Vasiljevic (2016).

67) AR: We have included few more lidar studies references. More particularly, in P2-3 L68-70 the following text has been added "It has been well established that the PPI method can also be applied to Doppler lidars (Cariou et al., 2007; Vasiljević et al., 2016) with the possibility to compute the mean wind profile by using a modified version of the VAD method as it has been demonstrated in the studies of Banta et al., (2002) and Chai et al., (2004).".

68) RC: Page 3 line 72. Replace "by eye" by "visual inspection" or similar.

68) AR: In P3 L77 the phrase "visual observation of the scans" has replaced the initial "by eye".

515

69) RC: Page 3 line 73. More than time-consuming, it might be non-systematic.

69) AR: In P3 L77-78 the sentence has been modified to: "However, the subjective classification by observing the images is a time-consuming approach and non-systematic.".

520 70) RC: Page 3 line 74. You meant "A less time-consuming" approach? What height was the met mast?

70) AR: The term less expensive refers to the sonic anemometers themselves without including the met mast. The data for the Barthlott et al (2007) study was taken at a 30 m tower.

71) RC: Page 3 line 78. "This study aims to identify turbulent coherent structures from single Doppler lidar horizontal scans".
 525 Also, please introduce here what is texture analysis (roughly maybe) and what what machine learning technique you are using.

71) AR: Regarding the texture analysis, we have added a brief description. See also response (16) to the first referee. For the machine learning technique, we also added a brief description. See also response (55) to the second referee.

72) RC: Page 3 line 83. Section 0 must read section 3, here and in the rest of the text.

530 72) AR: In P3 L96 Section 3 has replaced the initial Section 0 and it has been corrected throughout the text.

73) RC: Page 3 line 86-91. It should read "measurement campaign". Move "in Paris" to the end of the sentence, modified to "in the urban area of Paris". Remove the url of leosphere to the reference section maybe. More than only sensitive to the radial component, the lidar does measure and it is intended to measure only the radial component. Lidars technology and its operation principle is well known, use references (Cariou, maybe write ts paragraph in amore concise way, being specific in the corresponding table than in the text here, which is a bit confusing.

73) AR: AR: In P3 L100, the phrase "A two-month measurement campaign" has replaced the initial "A two-months campaign". In P3 L102 the phrase "in the urban area of Paris" has been inserted. The Cariou (2007) reference has replaced the initial "Kumer et al. (2014) and Veselovskii et al. (2016)" in P4 L109.

540

74) RC: Page 3 line 101. I would say fast instead of short. Also, what type of structures and why this time window?

74) AR: We refer to coherent structures, in P4 L115 the sentence "The duration of each scan was 3 minutes which is sufficiently fast for the observation of coherent structures with a lifespan of several minutes." Has replaced the initial "The duration of each scan was 3 minutes which is sufficiently short for the observation of structures."

545 The time-window of 3 min is the result of the selection of the 2° azimuth angle resolution. We wanted to combine a high spatial resolution with a time-window that would allow us to observe coherent structures.

75) RC: Page 4 line 121. What is the reason behind a is small for your case?.

75) AR: The parameter *a* is associated to the vertical wind speed, more particularly with the parameter  $w\cos\phi$  of the radial 550 wind speed analysed in its' wind components for a line of sight beam with azimuth angle  $\theta$  and elevation angle  $\phi$ :

 $Vr = u \cos\theta \sin\phi + v \sin\theta \sin\phi + w\cos\phi$ , where u is the longitudinal, v is the transverse

and *w* is the vertical wind component respectively. The offset is associated to the parameter  $wcos\phi$  (Thubois et al. 2018, Study of the configurations and scanning strategies of Doppler Lidars for providing wind and aerosol/cloud profiles).

In our case the elevation angle is only 1° thus a is around 10 times smaller than the parameter b throughout our data.

555

76) RC: Page 5 Figure 2. Why the reach of the lidar is 2 km and no 5 km?.

76) AR: Due to the CNR

We only used night streaks because the wind shear is a clear indication for the existence of streaks. As the algorithm only uses the five texture analysis parameters for the classification (Figure 8 of the manuscript), it shouldn't affect the results.

**590**

81) RC: Page 6 line 166. How many cases did you use for the "Other" category? From table 5 seems that they are around 60, the double. What is the reason for such big number?. Can this influence the final classification output? This is explained in section 4, but it should be clear from here.

81) AR: We used 60 cases which is double compared to the rest. The reason is that some of the machine learning algorithms 595 are sensitive to the balance of classes in the training data (see Kubat 2017, p 194). If one category is dominant but for the training ensemble all categories are represented by the same number of cases, then the algorithm can overestimate or underestimate a category. Nevertheless, we also tried a training ensemble with all the categories represented by 30 cases and the results were similar.

600 82) RC: Page 7 Figure 4. What is the scale of map in (d) and (e)?

82) AR: We have added the scanning area in Figures 5d and 5e of the manuscript.

83) RC: Page 7 line 176. Could you introduce what texture analysis is first? Additionally, since "Others" had a poor fitting and then uncertain wind direction, how did you align them with 0 degrees?

605 83) AR: We have included a bried description of texture analysis as showcased in response (16) to the first referee. The VAD method was used even for the bad cases, as the radial wind field in this case fell in the category of not interesting. As it can be seen in Figure 4 of the supplementary materials, it is still possible to fit a cosine function even for a bad fit and but the patterns are not interesting as illustrated in Figure 3 of the supplementary material.

610 84) RC: Page 7 line 180. Eight bins were chosen for increased contrast. Why eight?, could you develop more on this?. What is the effect of the number of bins in the output?

84) AR: The scope was to enhance the contrast of the structures for a better visualization of the alternating positive and negative areas in the radial turbulent wind field as we explained analytically in response (17) to the first referee.

615 85) RC: Page 8 line 185. The procedure for the construction of the CM matrix is a bit confusing. Could you write it in a more concise way?

85) AR: In P9 L226-227 the following sentence "The rows and columns of the CM represent the wind levels from 1 to 8, whereas the cells contain the frequency of the combination of two neighbour pixels in the image" has replaced the initial "The rows and columns of the CM represent the wind levels from 1 to 8, whereas the cells contain the number of occurrences of neighbour pairs with velves corresponding to the row and column index."

620 neighbour pairs with values corresponding to the row and column index.".

86) RC: Page 9 line 212. Is it possible to elaborate more on the 4 parameters described?. It is not clear only from the equations what their characteristics are.

- 86) AR: The following text has been added in the manuscript in P10 L251-255: "Correlation indicates the existence of linear structures in the image, with high values associated to a large amount of linear structure in the image. Contrast reveals the local variations in an image, where a large amount of variations leads to high values. Homogeneity is self-explanatory and the high values represent a homogeneous image. Finally, energy measures the uniformity of an image with the highest values corresponding to constant or periodic forms (Haralick et al., 1973; Yang et al., 2012).".
- 630 87) RC: Page 10 Figure 6. The notation of the azimuth angle is different form the text. Why does homogeneity grow after 45 degrees for all categories? The definition of homogeneity says that CMs with large values in the diagonal might result in larger values of this parameter. The diagonal from table 3 to 4 decrease because of azimuth angle. Should homogeneity decrease monotonically from 0 to +/- 90 degrees?. Can you elaborate more on this?. How many cases are represented for each category in the figure? Only one scan? An average from many cases?
- 635 87) AR: The notation of the azimuth angle in the y axis of the Figure 7 in P11 has been corrected. The angle also represents the distance between two grid point. For 45° angles or above, the distance between two grid points are n rows whereas below 45° they are n-1, n-2 etc. We have included an ideal case in Figure 5 of the supplementary material. Regarding whether the homogeneity should decrease monotonically from 0 to +/- 90 degrees, that depends on the case and on the order of the neighbour. When we have elongated patterns then yes we can see the prominent peak at 0° as we see in Figure 7 for the streaks
- 640 and in some degree for rolls but not so much for thermals and others. In the ideal case of Figure 5 in the supplementary material it is also possible to see how the values of the co-occurrence matrix change according to the periodicity of the patterns.

**88) RC: Page 10 line 231. Notation is a bit weird here.**

88) AR: The maximum and minimum refer to the azimuth angle  $\varphi$ . We have modify Equation 8 in P11 of the manuscript as follows:

Hom.  $Amp(n) = max_{\varphi}(Hom(\varphi, n)) - max_{\varphi}(Hom(\varphi, n))$

89) RC: Page 10 line 241. The description of the training set might be better place in section 3. Why is it expected that "other" category should double the rest? Please elaborate more on this.

89) AR: The preliminary analysis of patterns showed that "others" class is approximately twice abounded than each other class. We decided to double the number of examples "others" class in the training dataset, as some of machine learning algorithms are sensitive to the balance of classes in the training data (see Kubat 2017, p 194).

90) RC: Page 11 Figure 7. This figure is very confusing and not self-explanatory at all. Please give more information in its caption, relative to the number in parenthesis (neighbor order I suppose), state that they are all or a few of the final parameters used.

90) AR: In P 13 Figure 8 the following caption has been added: "Parameters selected to minimize the classification error of the training ensemble by the QDA method. From left to right: Amplitude of the homogeneity for the 2nd-neighbour, integral of the contrast for the 18th-neighbour, amplitude of the contrast for the 4th-neighbour, integral of the correlation of the 8th-neighbour and symmetry of the homogeneity for the 2ndneighbour." has replace the initial "Figure 7: Texture analysis

660 parameters selected to minimize the classification error of the training ensemble by the QDA method.".

655

91) RC: Page 12 Table 5. Change "eye-made" to a better term, like visual classification or similar.

91) AR: In P13 the word "visual" has replaced the initial "eye-made" in the caption of Table 5.

665 92) RC: Page 12 line 286. It is not clear if streaks were also detected during daytime, since the previous definition of the training set (line 162) says only night-time, but figure 9 says the opposite. Same for rolls and thermals. In summary, the constraint you talk about (day-time rolls and thermals, night-time streaks) does concern only the training set definition?

92) AR: Up until Figure 8 of the manuscript we showed the results for the training ensemble, where we only considered rolls and thermals during day and streaks during the night. So the classification error of the algorithm in Figure 8 refers to the training ensemble. The algorithm was able to classify our training ensemble successfully for approximately 91% for the texture analysis parameters of Figure 8. Then we use these parameters to classify all the 4577 scans and the results are presented in Figures 9 and 10 of the manuscript, thus we detect streaks during the day and thermals and rolls during the night.

- 93) RC: Page 12 line 292. If I am correct, you tried to explain thermals during night, not others during days. Only the last word in the sentence, "reverse", explains this. Moreover, can you elaborate more on what is the reason behind the erroneous classification of thermal as "others"? During stable conditions turbulent eddies are smaller, structures also show smaller length-scales. However, mean wind can show slight differences with no directional preference, and they can look like thermals (see Shah and Bou-Zeid, 2014).
- 93) AR: The category "others" includes the patterns that cannot be classified to one of the three turbulent structures type. It is
  possible to have a not symmetrical wind field, thus a bad case, during the day as well. In fact, 10 bad cases of the training ensemble for the category "others" occurred during the day. The physics behind the misclassification are very interesting. However, in our case the misclassification is linked to the shape of the patterns. It is possible that another texture analysis parameter could improve the distinction between these two types.
- 685 94) RC: Page 13 line 295. Stable cases during night show buoyant forces opposing vertical momentum flux and turbulence generation. Mechanical turbulence does die out under stable conditions. Mechanical turbulence destruction by buoyancy is the dominant mechanism, not the opposite.

94) AR: In P14 L360-362 we have added the following sentence "This was also expected as the nocturnal low level jets is a main driving factor for the formation of streaks and we observed the occurrence of wind shear higher than 2 s-1 over Paris for 20 out of the 62 nights during the VEGILOT campaign.".

95) *RC*: Page 13 line 314. So thermals are not turbulent?. Why do you separate rolls and steaks form thermals? Does it has to do with pollution transport or something similar?

95) AR: Rolls and streaks are the focus on many boundary layer studies. In the specific sentence, we wanted to emphasize the regularity of observing coherent structures over Paris during the period of our study. Moreover, in the study we are currently working on with regards to the physics of the structures, the transition between the structures for particular cases (e.g. low level jets, cloud streets etc.) will be one of the focal points.

**List with all the changes**

700

In P1 L8 the affiliation "Institute of Numerical Mathematics, Russian Academy of Sciences, Moscow Russia" has changed to "Marchuk Institute of Numerical Mathematics Russian Academy of Sciences, Moscow, Russia".

In P1 L12-L21, P3 L96-L110, P4 L127-L142, P5 L165, P6 L176, P7 L188-L189-L190-L193-196, P8 L225, P9 L231, P9 L238, P9 L241, P12 L297-L314, P14 L373-L374, P15 L391, P16 L417-L420 the term "Turbulent structures" has changed to "Medium-to-large fluctuations and coherent structures (mlf-cs)".

In P1 L13 the word "manually" has changed to "visually".

In P1 L14-L20-L28, P2 L70, P3 L85-L86-L92-L94, P14 L374, P16 L414-L418 the word "turbulent" has been removed.

In P1 L16 the word "and" has been removed.

In P1 L17-18 the sentence "The lidar recorded ..... using the velocity azimuth display method" has been replaced by the sentence "The mlf-cs of the radial wind speed ..... over 4577 quasi-horizontal scans".

In P1 L24 the phrase "namely the" has been added before "quadratic discriminate analysis" which is now without a parenthesis.

In P1 L24-25 the sentence "Using the 10 fold cross validation ..... in particular for streaks" has been replaced by the text "The algorithm was able .... classification error equivalent to 3.3%".

In P2 L32-35 the paragraph "The understanding of the connection .... low turbulence (Kallos et al, 1993)." has been removed.

715 In P2 L37 the phrase "instantaneously space and phase correlated" has changed to "maintenance of the phase-averaged".

In P2 L39-40 the sentence "Furthermore, a coherent ..... averaged statistics calculations (Hussain, 1983)" has been removed.

In P2 L42-49 the text "Several studies have been ..... out of the scope of this study." has been added.

In P2 L50-51 the sentence "Turbulent streaks are structures .... with an ascendance (Khanna and Brasseur, 1998)." was moved to the beginning of the paragraph.

720 In P2 L59 the reference "Lemone, 1972" changed to "LeMone, 1972".

In P3 L79-81 the sentence "The PPI method ...Doppler lidars." changed to "It has been well .... in the studies of Banta et al., (2002) and Chai et al., (2004)."

In P3 L87-89 the sentence "They combined a two dimensional ..... of the scans by eye." changed to "They combined quantitative characteristics ..... visual observation of the scans.".

**725 In P3 L90 the phrase "and non-systematic" has been added following the phrase "time-consuming approach".**

In P3 L94-95 the sentence "However, their study ..... observe via a lidar" has been added.

In P3 L97-98 the phrase "based on the combination .... Quadratic Discriminate Analysis (QDA)" has been added.

In P3 L99-107 the text "Texture analysis is an effective .... and unstable conditions." has been added.

In P3 L107 the word "It" changed to "The classification method".

730 In P3 L108 the expression "2-month" changed to "two-month".

In P3 L110 "Section 0" changed to "Section 3".

In P4 L114 the expression "A two-months campaign" changed to "A two-month measurement campaign" & "in Paris" has been removed.

In P4 L116 the phrase "in the urban area of Paris" has been added.

**T35 In P4 L124 the references "Kumer et al. (2014) and Veselovskii et al. (2016)" changed to "Cariou et al., 2007)".**

In P4 L124 the phrase "The lidar is sensitive only to" changed to "A wind lidar is measuring".

In P4 L126 the phrase "scanning sequence as it was implemented" changed to "implemented scanning methods".

In P4 L129 the word "for" changed to "with".

In P4 L130 the word "short" changed to "fast" & the phrase "observation of structures" changed to "observation of coherent 740 structures with a lifespan of several minutes".

In P4 L132-136 the text "It is noteworthy that .... homogeneous urban surface" has been added.

In P4 L136 the verb "rise" changed to "was risen".

In P4 L137-141 the text "It was also important .... approximately 15 seconds" has been added.

In P5 L150-153 Table 2 changed to showcase only the scanning methods PPI and DBS that was used for this study & the 745 caption "Scanning sequence during VEGILOT" changed to "Scanning methods selected during VEGILOT". Additionally, the elevation of "90°" has been removed from the DBS properties as it was not selected for the experiment.

In P5 L153 the "Figure 2" has been added to showcase the ground altitude map for the scanning area. As a result, the numbering of the Figures has been modified accordingly throughout the manuscript.

In P5 L166-169 the text "A pareameter that indicates .... 3 km." has been added.

750 In P6 L172 the "Equation 2" has been added along with the text "and it was computed ..... f is the fitted curve.". As a result, the numbering of the Equations has been modified accordingly throughout the manuscript.

In P6 Figure 3d & in P8 Figures 5a, 5b, 5c & in Figures 6a, 6b the title of the colorbar "turbulent radial wind field" changed to "Mlf-cs field".

In P6 L178 the word "The" has been added in the beginning of the paragraph.

755 In P6 L179 the expression "homogeneity of the wind field" changed to "regional wind flow".

In P6 L179-181 the sentence "Troude et al ..... for the local wind speed" has been added.

In P7 L192 the words "asceding" and "descending" have been switched.

In P7 L199 "Sec. 4.1" changed to "Section 4.1".

In P7 L206-207 the sentence "It is important to note ..... lower part of the rolls" has been added.

760 In P7 L209 the phrase "signifying fair weather conditions" has been added.

In P7 L214 the units "m/s" changed to "s-1".

In P7 L215-217 the formula for the horizontal wind speed computed from the DBS observations has been added as Equation 3 accompanied by the sentence "The horizontal wind speed ..... via the expression.". As a result, the numbering of the Equations has been modified accordingly throughout the manuscript.

765 In P8 L217-219 the sentence "For many cases ..... generation of turbulence" changed to "Consequently, the wind shear .... were selected.".

In P8 L223-224 the sentence "Regarding rolls .... VAD method was applicable." has been added.

In P8 L227 the expression "turbulence pattern type" changed to "structure type".

In P9 L243 the phrase "number of occurrences of neighbour pairs with values corresponding to the row and column index" 770 changed to "frequency of the combination of two neighbour pixels in the image".

In P10 L268-271 the text "Correlation indicates the existence of linear structures in the image, with high values associated to a large amount of linear structure in the image. Contrast reveals the local variations in an image, where a large amount of variations leads to high values. Homogeneity is self-explanatory and the high values represent a homogeneous image. Finally, energy measures the uniformity of an image with the highest values corresponding to constant or periodic forms (Haralick et al., 1973; Yang et al., 2012)." has been added.

775

In P11 Equation 8 the notation " $\varphi$ " has been added.

- In P12 L298 the expression "The Quadratic Discriminant Analysis" changed to "The QDA".
- In P12 L300-311 the text "QDA or normal Bayesian ..... then the process stops." has been added.
- In P12 L315-316 the text "The algorithm can be ..... was preferred (Kubat, 2017)." has been added.

780 In P12-13 L328-341 the text "In particular, these parameters .... for chaotic ones ("others")" changed to "Analytically, these parameters ..... horizontal sizes differ.".

In P13 L347-348 the sentence "The significant parameters ..... with different sizes" changed to "Furthermore it demonstrates .... with different sizes.".

In P13 L351-354 the caption of Figure 8 "Texture analysis parameters ..... by the QDA method" changed to "Parameters selected ..... for the 2nd neighbour.".

In P13 L356 the sentence "The algorithm allowed classifying correctly about 91% of the training ensemble." has been added.

In P14 L364 the "eye made" changed to "visual".

In P14-15 the text "It is evident that despite time was a much less significant ..... classified at night." changed to "It is evident that despite time was not one of the selected .... classified at night."

790 In P15 L386-387 the sentence "This was also expected ..... formation of streaks" changed to "This was also expected ..... during the VEGILOT campaign.".

In P15 L391-393 the sentence "The current study ..... single Doppler lidar" changed to "The current study ..... supervised machine learning technique".

In P15 L393-394 the sentence "More particularly a two months ..... 1° elevation" has been removed.

795 In P15 L394-399 the text "The VAD method was used ... enclosed patterns (unaligned thermals)" changed to "By applying the VAD method .... enclosed patterns (unaligned thermals)".

In P15 L400-401 the sentence "In order to apply ..... and unaligned thermals" changed to "A training ensemble .... And unaligned thermals.".

In P15 L402 "For this purpose" changed to "Subsequently".

800 In P15 L403 the phrase "to the training ensemble in" has been added & "and" has been removed.

In P15 L404 the sentence "The algorithm allowed ..... analysis parameters as predictors." changed to "The results showed ..... analysis parameters as predictors."

In P15-16 L405-414 the text "More particularly, these parameters ..... such as streaks and rolls" has been added.

In P16 L428-L430 the initials HD have been added to the authors' contribution.

805 In P16 L445-446 the acknowledgement "Experiments presented in this paper ..... supported by SCoSI/ULCO (Service COmmun du Système d'Information de l'Université du Littoral Côte d'Opale)." has been added.

In P16 L447 the acknowledgement "This study was funded by RFBR, project number 20-07-00370." has been added.

In P17 L449-457 the references of "Adrian, 2007, Alparone et al., 1990, Aouizerats et al., 2011, Banta et al., 2002" have been added.

810 In P17 L460-461 the reference of "Bonament,e 2017" has been added.

In P17 L466-472 the references of "Cariou et al., 2007, Castellano et al., 2004, Chai et al., 2004" have been added.

In P17 L488-490 the reference of "Holli et al., 2010" has been added.

In P18 L500 the reference of "Kallos et al., 1993" has been removed.

In P18 L500-504 the references of "James et al., 2000, Kayitakire et al., 2006" have been added.

815 In P18 L513 the reference of "Kumer et al., 2014" has been removed.

In P18 L513 the reference of "Kubat, 2017" has been added.

In P18 L518 the "LeMone" reference has been corrected.

In P18 L520 the reference of "Lemonsu and Masson, 2002" has been added.

In P18 L531 the reference of "Roth, 2007" has been removed.

820 In P18-19 the references of "Saint-Pierre et al., 2010, Sandeepan et al., 2013" have been added.

In P19 L538 the reference of "Soldati, 2005" has been added.

In P19 L545 the reference of "Troude et al., 2002" has been added.

In P19 L549 the reference of "Veselovskii et al., 2016" has been removed.

In P19 L549 the reference of "Vasiljević et al., 2016" has been added.

[revised manuscript text omitted]

---

## Author Response (AR2)

**Authors' response to the first referee**

*1) Referee Comment: The altitude variations that the elevation map shows in Figure 2 in combination with the height variations of the buildings increase the heterogeneity of the terrain. I suggest explicitly stating that for the needs of the study the flow over the scanned areas is assumed to be homogeneous and avoid characterising the terrain as homogeneous.*

1) Authors' response: In order to avoid any confusion for the readers, we have removed the term homogeneous terrain or homogeneous urban surface from the text. We instead only refer to the assumption of a homogeneous wind field within the scanning area.

*2) RC: The root mean square error is an index on the deviations between the data sample and the corresponding predicted values of a model fit. Therefore, it is a way to evaluate the quality of the sinusoidal fit. The authors argue that instead they used the "symmetry of the radial wind field" to separate the bad cases. However, they do not explain how they have performed this step. Was it based on a visual inspection or on a statistical quantity? Please add more information in the manuscript regarding this step.*

2) AR: We have separated the good and the bad cases of the training ensemble based on the symmetry of the radial wind fields by visually examining the radial wind fields and their individual cosine function fits. This information is now added on the text in P7 L203-205.

3) *RC: The authors explain that: "the wind shear was estimated from the vertical profile of V_hor by subtracting the local minima from the local maxima above it, near the surface." How consistent was the location of the local minima and the maxima in the data set used? I suggest to add one sentence stating this information. Furthermore, it should be added that the wind speed difference was divided by the corresponding difference in height between the local minima and local maxima.*

3) AR: The location of the local minima and local maxima was rather consistent. The local maxima ranged between 200 and 300m with the local minima located approximately 200m above it. Regarding the second part of your comment related to the definition of the wind shear, we realized that in our previous response and in the submission of the manuscript we falsely used the term wind shear. We actually followed the definition by Stull (1988) that a llj is characterized by a maxima local wind speed more than 2 m/s higher than the wind speed above it. Therefore, in P7 of the manuscript we have now replaced the term wind shear with the definition of local maxima and minima according to Stull. The units have been also corrected to 2m/s instead of $2^{s-1}$.

*4) RC: Here the authors describe their experience regarding the optimal contrast used to highlight the coherent patterns. This experience could be of value to others that would be interested in performing a similar analysis. The main points of their answer can be added as a discussion before the conclusions of this article.*

4) AR: Thank you very much for your suggestion. The information regarding other binning options will be publicly available as our responses will be published if the paper will be accepted for final publication. We would rather prefer to present only the option with the best performance in the current paper and discuss the performance of other binning options, along with a possible optimization of the methodology that minimizes the error, in a separate in-depth study focusing on the algorithm.

**List with all the changes**

In P4 L120-123 the text “Furthermore, the ground altitude …. by a rather homogeneous urban surface” changed to “The ground altitude …. the wind field within the scanning area is homogeneous (see Section 3.1).”.

In P7 L195-198 the text “The observation of the horizontal wind profiles …. low level jet events (Stull, 1988) (Figure 5f).” changed to “The observation of the horizontal wind profiles…. ranging from 200 to 300 m and 400 to 500 m respectively.”

In P7 L200-201 the sentence “Consequently, the wind shear was ….. near the surface.” has been removed.

In P7 L202 the term “wind shear” has been replaced by the sentence “differences in local maxima and minima of the $U_{hor}$” and the units “$s^{-1}$” have been replaced by “m/s”.

In P7 L206-207 the sentence “The selection of symmetric radial wind fields …. individual cosine function fits.” has been added.

In P14 L369-370 the phrase “wind shear higher than 2 s-1” has been replaced by the phrase “the local maxima of the horizontal wind speed near the surface higher than 2 m/s compared to the local minima”

In P16-19 L420-533 the references have been modified to comply with the guidelines of the AMT journal. The year of the publications has been moved to the end of the reference, capital or small letters and punctuations have been corrected.

In P16 L422-424 the proper conference name has been added “Radar and Signal Processing, IET Digital Library”.

In P16 L433 the name of the chapter has been added “Functions of random variables and error propagation”.

In P16 L434, P17 L462, P18 L489 the name of the publisher and location have been added “Sringer, New York, USA”.

In P17 L442 the numbers of the volume and pages of the paper have been added “59, 1061-1069”.

In P17 L445 the term “American Meteorological Society” has been removed.

In P17 L459-460 the correct journal has been added “IEEE Transactions on systems man and cybernetics”.

In P17 L475-476 the journal has been corrected from from “Current medicinal chemistry” to “Springer Texts in Statistics, Springer, New York, USA”.

In P18 L488 the duplicate “An Introduction to Machine Learning” has been removed.

In P18 L498 the DOI of the paper has been added “https://doi.org/10.1175/1520-0469(1962)019%3C0343:NOWVWD%3E2.0.CO;2”.

In P18 L514 the “JTECH-D-19-0120.1” has been removed.

In P18 L516 the name of the journal has been added “J. Appl. Math. Mech.”.

In P18 L520-521 the name of the publisher and location as well as the DOI of the book have been added “Springer, Dordrecht, Germany, https://doi.org/10.1007/978-94-009-3027-8”.

In P18 L523 the numbers of the pages of the paper have been added “1-25”.

In P19 L534 the numbers of the volume and pages of the paper have been added “134, 5-22”.

In P19 L536 the name of the report has been added “NOAA Technical Memorandum ERL ARL-61, 83”.

In P19 L541 the numbers of the volume and pages of the paper have been added “83, 997-1002”.

[revised manuscript text omitted]